# IVCR-200K: A Large-Scale Multi-turn Dialogue Benchmark for Interactive Video Corpus Retrieval

## Abstract

In recent years, significant developments have been made in both video retrieval and video moment retrieval tasks, which respectively retrieve complete videos or moments for a given text query. These advancements have greatly improved user satisfaction during the search process. However, previous work has failed to establish meaningful **"interaction"** between the retrieval system and the user, and its one-way retrieval paradigm can no longer fully meet the personalization and dynamics needs of at least 80.8% of users.

In this paper, we introduce a more realistic setting, the Interactive Video Corpus Retrieval task (IVCR) that enables multi-turn, conversational, realistic interactions between the user and the retrieval system. To facilitate research on this challenging task, we introduce IVCR-200K, a bilingual, multi-turn, conversational, abstract semantic high-quality dataset that supports video retrieval and even moment retrieval. Furthermore, we propose a comprehensive framework based on multi-modal large language models (MLLMs) to support users' several interaction modes with more explainable solutions. Our extensive experiments demonstrate the effectiveness of our dataset and framework. The datasets, codes and leaderboards are available at: https://ivcr200k.github.io/IVCR.

## 1 Introduction

With the rapid proliferation of video platforms such as YouTube and TikTok, an ever-increasing number of videos are being produced every day, underscoring the significance of the video retrieval task in the multi-modal field (Yan et al., 2023; Zhang et al., 2023; Zeng et al., 2021). Typically, users employ descriptive sentences, and the retrieval system (Xu et al., 2016; Luo et al., 2022) sorts by matching textual descriptions and visual videos, ultimately returning the user's preferred videos, as depicted in Figure 1(a). At a more granular level, as shown in Figure 1(b), researchers have proposed the video moment retrieval task (Gao et al., 2017; Zeng, 2022), which utilizes textual descriptions to retrieve a small moment within the complete video. These tasks significantly enhance user satisfaction during the search process.

However, the majority of video retrieval systems operate in a "one-way" manner, which may not fully cater to the personalized and dynamic preferences of users. This "one-way" approach inhibits user interaction with the system, resulting in every request from the user needing to be rewritten. In fact, it is a common phenomenon that users desire **"multi-turn interaction"** with systems. To delve deeper into this phenomenon, we devised a questionnaire[1] regarding user search behavior, depicted in Figure 2. A striking 80.8% of respondents expressed a preference for interactive search functionality. Similarly, within the ShareGPT[2] conversation dataset, the average interaction round between users and the chat system stands remarkably high at 7.27. Moreover, our questionnaire indicate that interactive demands exhibit intricate behavioral patterns, as illustrated in Figure 1(c): 1) Long2Short: Keep looking for clips within the long videos that have already been scanned. 2) Short2Long: Search full-length videos based on known short videos. 3) Analogous: When the user inputs "I would like to watch a movie similar to this clip", the system should be able to provide a video with similar content.

---

[1] Details of this questionnaire can be found in supplementary material A.
[2] https://sharegpt.com/

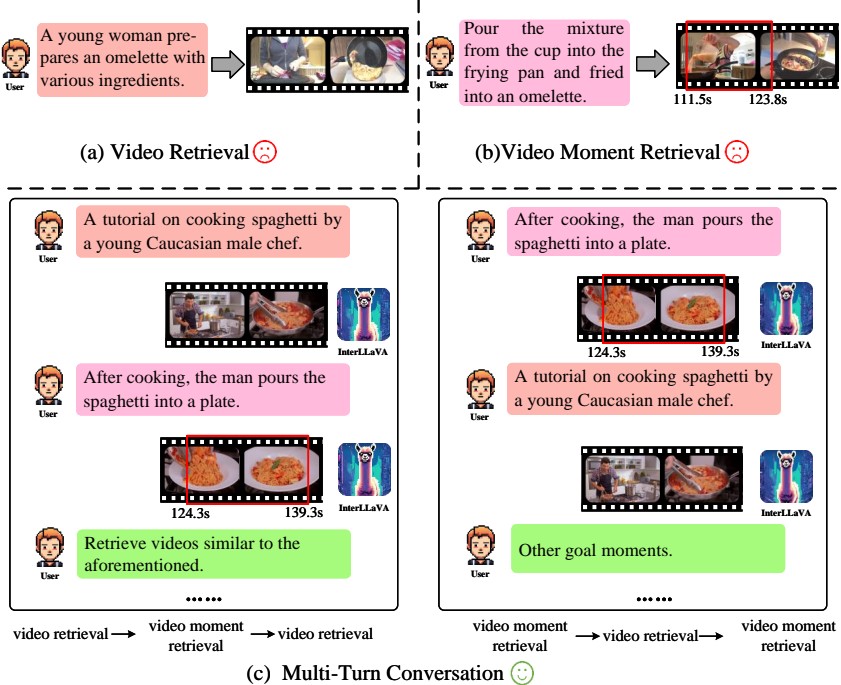

Figure 1: **Visualization of the video retrieval, moment retrieval and our interactive retrieval.**

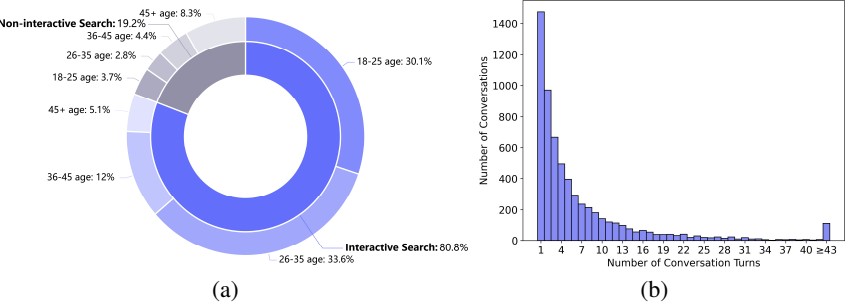

Figure 2: **Investigation of User Search Behavior Feedback and interaction turns in ShareGPT. Users demonstrate a pronounced inclination towards interactive search and harbor high expectations regarding interaction rounds.**

Therefore, drawing from these observations, we believe that the implementation of an interactive retrieval system holds significant value (Ma & Ngo, 2022; Maeoki et al., 2020), despite the challenges of complex user behaviors. Through multi-turn interaction with users, the system can adapt to individual preferences, furnishing more personalized retrieval outcomes. However, researchers have yet to delve deeply into this practical issue, one that resonates more closely with users' perspectives.

Formally, we introduce the Interactive Video Corpus Retrieval task (IVCR) for the first time. We define the "interactive" as meeting the following requirements: **1) Multi-turn.** This multi-turn interaction will extend the connection between the user and the search system. This process includes several interaction modes, such as video retrieval-only, moment retrieval-only, video-first-then-moment, moment-first-then-video, or creating a new topic for retrieval. **2) Free dialogue.** Users perform queries in natural language (Alayrac et al., 2022; Dai et al., 2024), and the retrieval system should explain the returned results in natural language form, which is more explainable and user-friendly. Furthermore, existing multi-modal retrieval datasets mostly contain low-level descriptive descriptions (e.g., "There are three dogs on the green lawn"), which do not align with the high-level abstract semantics used by users in real scenes (e.g., "Kung Fu movie where men and women fight"). **3) Real interaction.** The pioneers create simulated environments to generate interactive data (Ma

Table 1: Comparison of IVCR-200K and other existing video-language datasets.

| Dataset | Multi-turn | Dialogue | Real interaction | Videos | Queries | Language |
|---|---|---|---|---|---|---|
| MSR-VTT(Xu et al., 2016) | ✗ | ✗ | ✗ | 7,180 | 200K | English |
| MSVD(Chen & Dolan, 2011) | ✗ | ✗ | ✗ | 1,790 | 70K | English |
| LSMDC(Rohrbach et al., 2017) | ✗ | ✗ | ✗ | 200 | 118K | English |
| ActivityNet(Krishna et al., 2017) | ✗ | ✗ | ✗ | 20,000 | 100K | English |
| VATEX(Wang et al., 2019) | ✗ | ✗ | ✗ | 41,250 | 825K | English, Chinese |
| HowTo100M(Miech et al., 2019) | ✗ | ✗ | ✗ | 1.221M | 136M | English |
| Charades-STA(Gao et al., 2017) | ✗ | ✗ | ✗ | 6,670 | 16,128 | English |
| DiDeMo(Anne Hendricks et al., 2017) | ✗ | ✗ | ✗ | 10,464 | 41K | English |
| TVQA(Lei et al., 2018) | ✗ | ✔ | ✗ | 21,793 | 152,545 | English |
| AVSD(Alamri et al., 2019) | ✔ | ✔ | ✗ | 11,816 | 118,160 | English |
| **IVCR-200K (Ours)** | ✔ | ✔ | ✔ | 12,516 | 193,434 | English, Chinese |

& Ngo, 2022), but we emphasize that only truly understanding users can optimize a better search experience.

Unfortunately, at present, there is no available dataset or reliable framework to support this task of interactive video corpus retrieval, as shown in Table 1. **1) Dataset.** Existing video retrieval datasets are inadequate for multi-turn interaction scenarios, such as ActivityNet (Krishna et al., 2017) and DiDeMo (Anne Hendricks et al., 2017), which are single-turn datasets. Therefore, we propose an innovative interactive retrieval dataset, IVCR-200K, which is a bilingual, multi-turn, conversational, and abstract semantic high-quality dataset designed to support video retrieval and even moment retrieval. **2) Framework.** Existing retrieval methods are clearly insufficient for this conversational scenarios. For instance, solutions like CLIP (Luo et al., 2022; Fang et al., 2021) and 2D-TAN (Zhang et al., 2020) are discriminative models that cannot perform dialogue generation. Inspired by recent advances in multi-modal large language models (Li et al., 2023a; Ren et al., 2023), we combine their multi-turn dialogue, semantic understanding, and other capabilities to support users' interaction modes with a more explainable solution, named InterLLaVA. Extensive experiments demonstrate the effectiveness of our dataset and framework. We will release the code and dataset in the hope of contributing to the advance future research on real-world retrieval field.

The main contributions are summarized as follows: i)-To the best of our knowledge, this is the first work to introduce the "interactive" video corpus retrieval task (IVCR) , which effectively aligns users' multi-turn behavior in real-world scenarios and significantly enhances user experience. ii)-We introduce a dataset and an accompanying framework. Notably, the IVCR-200K dataset is a high-quality, bilingual, multi-turn, conversational, and abstract semantic dataset designed to support video and moment retrieval. The InterLLaVA framework leverages multi-modal large language models (MLLMs) to enable multi-turn dialogue experiences between users and the retrieval system.

## 2 RELATED WORK

**Video Retrieval Dataset**

In recent years, with the vigorous development of the digital video new media market and continuous technological innovation, the scale of datasets related to video retrieval has rapidly expanded. For example, Xu et al.(Xu et al., 2016) constructed a video understanding dataset MSR-VTT, which contains 10K clips and 20K different text descriptions corresponding to various categories. MSVD(Chen & Dolan, 2011) is also a dataset widely used in video retrieval, which contains 1,970 videos, and each video has approximately about 40 associated sentences. Rohrbach et al.(Rohrbach et al., 2017) built the LSMDC, with 200 movies and 128,118 sentences, which is widely used in cross-model retrieval between video and text. Krishna et al.(Krishna et al., 2017) built a large-scale dataset ActivityNet Captions for dense captioning events, which contains 20k videos and a total of 100k descriptions, each with its unique start and end times. In comparison, Howto100M(Miech et al., 2019) contains more than 23k different visual tasks and 136 million video clips from 1.22M instructional web videos with narration, which is the largest video retrieval dataset. Wang et al.(Wang et al., 2019) constructed a large-scale multilingual video description dataset VATEX, which contains over 41,250 videos along with 825,000 captions in both English and Chinese. Gao et al.(Gao et al., 2017) built a dataset

called Charades-STA, which augments the existing Charades (Sigurdsson et al., 2016) dataset by adding sentence temporal annotations for temporal activity localization via language. However, these datasets are mainly built to support video retrieval or video moment retrieval research rather than interactive video corpus retrieval, so they do not meet the personalized and dynamic retrieval needs of users. TVQA(Lei et al., 2018) is a large-scale video QA dataset based on six popular TV shows. It contains 152,545 QA pairs from 21,793 clips, spanning over 460 hours of video. AVSD(Alamri et al., 2019) is the only dataset for interactive video retrieval, which was created by adding dialogue data to the existing video dataset called Charades. Each video is associated with a 10-round dialogue discussing the content of the corresponding video. However, their annotations of 10-round dialogues are limited to each video, so they cannot be used for interactive video corpus retrieval.

In this paper, we built IVCR-200K dataset with 12K videos and more than 200K sentences covering 36 categories. To our best knowledge, IVCR-200K is the first and the largest video dataset for interactive video corpus retrieval. Dataset is a key step in developing deep learning based methods. We hope our dataset can inspire more efforts for the task of interactive video corpus retrieval.

**Video Retrieval.** Recently, numerous video datasets have been released for various video-language understanding tasks. In Table 1, we present a statistical comparison of our IVCR-200K dataset with ten video datasets used for video retrieval tasks. Video retrieval aims to retrieve relevant videos from a set of video candidates given a text query (Smeaton et al., 2006). Researchers have developed some pre-training systems (Luo et al., 2022; Fang et al., 2021; Gorti et al., 2022; Liu et al., 2022b). As an extension of video retrieval, video moment retrieval task aims to identify specific clips or moments within a video based on a given textual query (Gao et al., 2017; He et al., 2019). These studies have enhanced the service capabilities of the retrieval system. However, further development is required to meet the multi-turn interactive needs of users.

**Interactive Retrieval.** The concept of interactive retrieval has long been proposed in the context of combining human-machine learning techniques for multimedia content search (Thomee & Lew, 2012; Snoek et al., 2008). Currently, only a few works (Madasu et al., 2022; Maeoki et al., 2020; Ma & Ngo, 2022; Liang & Albanie, 2023) have explored this task. For example, Madasu et al.(Madasu et al., 2022) and Maeoki et al.(Maeoki et al., 2020)adopt a dialogue-based approach, utilizing a series of video-related questions and answers generated by different models as retrieval queries. Furthermore, Ma et al.(Ma & Ngo, 2022)develop a user simulation for intelligent multimedia applications to enable precise video segment search through human-computer interaction. The technical challenges in modeling multi-turn dialogue retrieval have contributed to the slow development in this direction.

**Large language Models.** With the breakthroughs in generative artificial intelligence, the way humans interact with machines has changed(Min et al., 2023; Zheng et al., 2024). Researchers have extended large language models to the visual perception domain, developing a series of large language models with multimodal information processing capabilities, such as Flamingo(Alayrac et al., 2022), BLIP-2(Li et al., 2023a), and LLaVA(Liu et al., 2024) for image processing, and Sora, Video LLaMA(Zhang et al., 2023), and Video Chat(Li et al., 2023b) for video understanding. Specifically, for interactive cross-modal video retrieval, future interactive video retrieval systems should function as "search assistants," engaging in genuine and coherent multi-round dialogues with users.

## 3 INTERACTIVE VIDEO CORPUS RETRIEVAL DATASET

### 3.1 DATASET COLLECTION AND ANNOTATION

To implement an interactive video retrieval system, we constructed a multi-turn, conversational dataset comprising 193,434 interactions sourced from 5 video repositories. This dataset encompasses functionalities such as video retrieval, video moment retrieval, and natural dialogue.

Illustrated in Figure 3, we devised a comprehensive collection pipeline:

- 1) Video source curation: Initially, we selected video datasets spanning diverse domains such as daily activities, movies, and kitchens, including selections like TVQA (Lei et al., 2018), LSMDC (Rohrbach et al., 2017),ActivityNet (Krishna et al., 2017), DiDeMo (Anne Hendricks et al., 2017), MSR-VTT (Xu et al., 2016), to ensure video source diversity. Subsequently, we filtered out select videos from these 5 original datasets. Videos featuring isolated actions or events, severe occlusion,

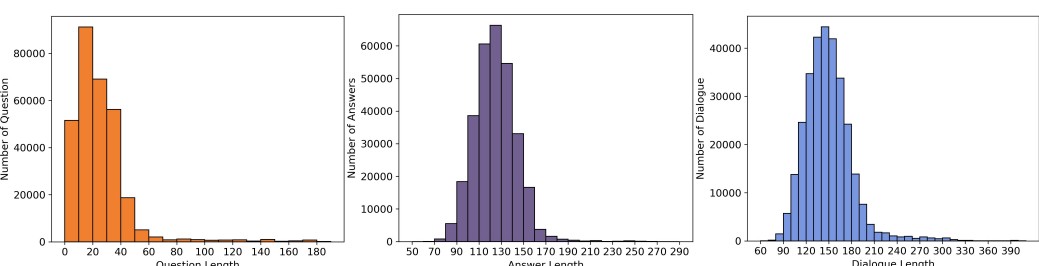

**Figure 3: The pipeline of our dataset collection.**

**Figure 4: Distribution of question lengths, answer lengths, and dialogue lengths.**

or excessively accelerated playback were excluded. Ultimately, 12,516 videos were chosen for inclusion.

- 2) Query refinement: Despite the presence of captions or descriptions with the filtered source videos, they often inadequately align with user queries in real-world scenarios. Hence, we employed GPT-4 for query refinement on captions. Specifically, we first combine the captions and user queries as input, and then use GPT-4 to generate user queries that more accurately and closely reflect the substantive content of the video.

- 3) Multi-turn dialogues: We established various dialogue dynamics, encompassing Long2Short, Short2Long, Long2Long, Short2Short, and Natural Dialogue scenarios. "Long2Short" denotes a user's inclination to pinpoint video clips further in the current round, while "Natural Dialogue" reflects users perceiving our system as a standard chat robot. Notably, while most dialogues consist of concatenated single-round exchanges, we also gathered a limited number of multi-turn dialogues from actual users.

- 4) Interpretability: To bolster the interpretability of interactive retrieval systems, we utilized GPT-4 to craft responses, encompassing intent understanding, retrieval or localization results, and reasons.

- 5) Bilingual capability: To broaden the reach of this dataset, we employed a translation model to render the dataset into Chinese.

Notably, every output produced by GPT-4 will undergo meticulous scrutiny and refinement by human experts to guarantee the precision of knowledge. Additionally, we implemented a validation process conducted by a review team, focusing on the quality and consistency of annotations provided by different annotators. After all annotations (193,434 sentence-level queries) were completed, the reviewers further examined the annotated data. Ultimately, we acquired a multi-turn, conversational dataset comprising 200K volumes, named IVCR-200K. The entire annotation and review process took approximately five months. More details on annotation procedure is provided in the supplementary material C.

### 3.2 DATASET ANALYSIS.

**Property Quality.** The statistical analysis of the property quality for video and textual query in the IVCR-200K dataset is shown in Figure 4 and Figure 5. In Figure 4, we present the length distribution of questions, answers, and dialogues within IVCR-200K. The average length of questions and answers in IVCR-200K is 24.5 words and 124.2 words, respectively. In contrast, the average length of questions in AVSD(Alamri et al., 2019) is 7.9 words, and the average answer length is 9.4 words. This indicates that the dialogues in our dataset are more verbose and conversational.

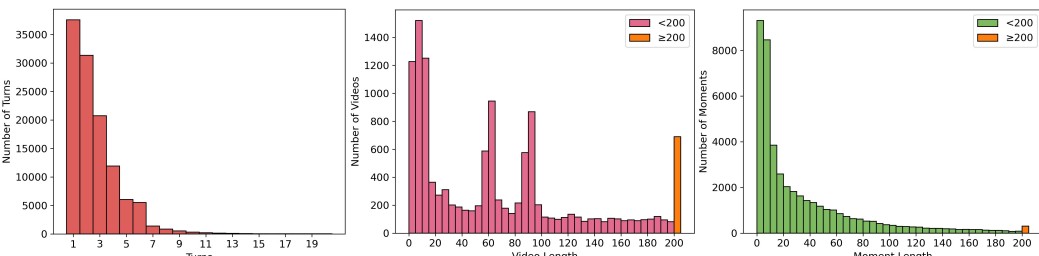

**Figure 5: Distribution of turn lengths, video lengths, and moment lengths.**

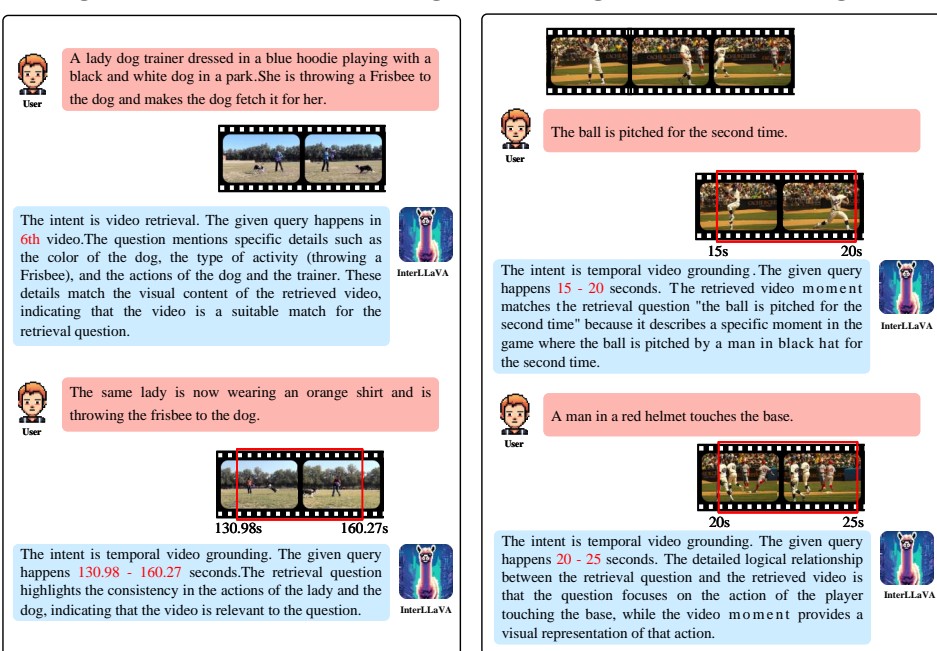

**Figure 6: Examples from the IVCR-200K dataset.**

Additionally, Figure 5 shows the distribution of the number of turns in multi-turn dialogues. The total number of dialogue turns is 302,074, with an average of approximately 2.6 turns, which aligns with typical user retrieval behavior. Figure 5 also presents the length distribution of videos and video moment. The average length of videos is 67.26 seconds, and the average length of video moments is 34.81 seconds, with most video moments being under 60 seconds.

**Diversity Quality.** We conducted an analysis of our video sources, the different types of videos, and performed a frequency analysis of annotated sentences, as detailed in supplementary material C.

**Visualization Quality.** We also check some cases as shown in the Figure 6. More examples are available in the supplementary material D.

## 4 INTERACTIVE VIDEO CORPUS RETRIEVAL FRAMEWORK

### 4.1 TASK DEFINITION.

Let $u_{(\cdot)}$ denotes a user whose historical interactive sequence is $Q = \{q_1, q_2, q_3, q_4, ...\}$, where $q_{(\cdot)}$ represents different textual queries. Formally, the goal of this interactive video corpus retrieval task is to retrieve semantically matched videos or moments in each round $i$, based on historical interactive sequence $Q_{<i}$. Among them, video moment retrieval requires not only the prediction of the most suitable video $v_j$, but also the prediction of the optimal moment within $v_j$, which includes the start $s$

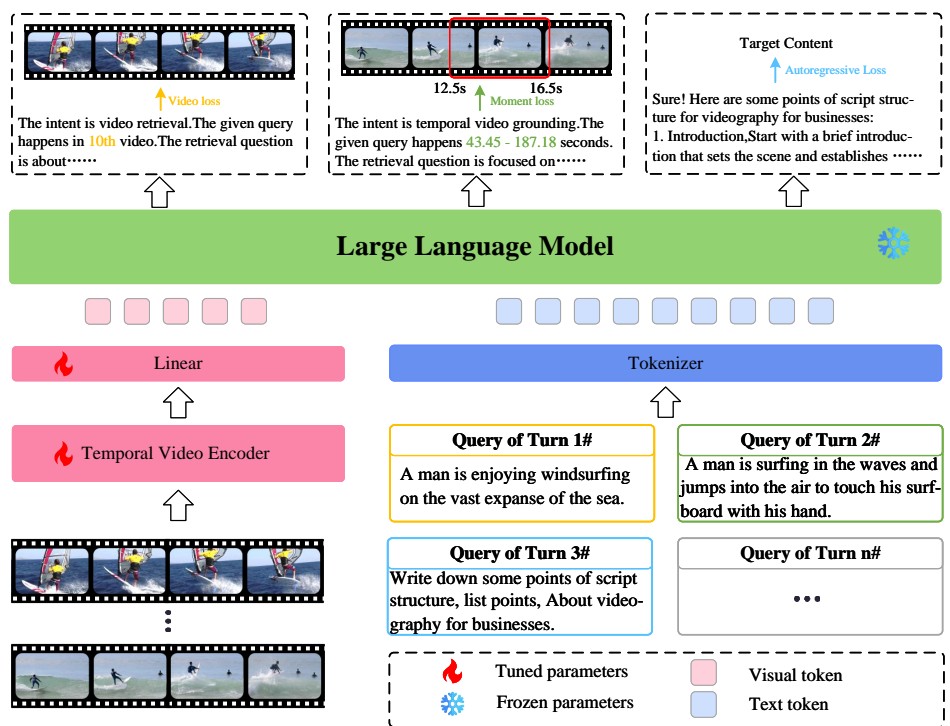

Figure 7: An overview of our framework for interactive retrieval.

and end $e$ timestamps. In addition, the interactive video corpus retrieval task is not limited to video retrieval, but also specifically considers the identification and processing of natural dialogue intent.

## 4.2 TASK PROCESSING.

As illustrated in Figure 7, our InterLLaVA adapts the pretrained multi-modal large language model LLaMA-2 (7B)(Touvron et al., 2023) to tackle video retrieval, video moment retrieval, and natural dialogue in a multi-turn setting. It takes video and text query as inputs and outputs video, video moment, and natural dialogue related to textual query intent, while providing interpretable feedback. Specifically, we fine-tuned Inter-LLaVA using instruction-tuning data, which generally consists of video-instruction pairs. Here is an illustrative example, with the underlined part serving a pseudo-instruction:

---

Video Retrieval:
Question: ### Human: [User Query] <VID> <Video Start> [Video Tokens] <Video End> [Instruction]
Answer: ### Assistant: The intent is video retrieval. The given query happens in <VID> video. [Explainable Feedback]

Video Moment Retrieval:
Question: ### Human: <Video Start> [Video Tokens] <Video End> [Timestamps] [User Query]
Answer: ### Assistant: The intent is temporal video grounding. The given query happens in [Start Time] - [End Time] seconds. [Explainable Feedback]

---

During the instruction fine-tuning of InterLLaVA, text query is first performed using a pre-trained multi-modal large language model (LLaMA-2 (7B)), which is then concatenated with video and answer prompts to serve as the input for InterLLaVA. The answer prompts include retrieval intent, video/moment cues, and interpretable feedback. Later, the answer prompts are utilized as the "ground truth" of InterLLaVA's generation. In the following, we elaborate the implementations of the three tasks.

**Video Retrieval.** For this task, we propose combining a fast two-tower video model with a multi-modal large language model through a re-ranking mechanism. Specifically, in the first phase, the video retrieval model predicts the top-10 video sequence $V_j$ based on videos and text queries. In the second phase, these top-10 video sequences and the text queries are input into a multi-modal large language model for re-ranking, outputting the most relevant video $v_j$. This approach retrieves the most relevant videos efficiently, reduces the memory and computational burden on the language model, and excludes irrelevant content. Notice that the first phase adopts offline video sequence extraction, while the second phase is trained end-to-end with the other tasks.

**Video Moment Retrieval.** For this task, we employ a traditional two-stage retrieval method, utilizing a fast two-tower model for video retrieval and a multi-modal large language model for precise moment localization. Specifically, we implement a two-phase approach. In the first phase, the video retrieval model directly output the top-1 video $v_j$. In the second phase, the textual query and the top-1 video are input into a multi-modal large language model to generate reasonable and coherent response and video moment. To enhance the feature fusion in the time dimension, we adopt a sliding video Q-Former and initialize it from the Video-LLaMA(Zhang et al., 2023) checkpoint. Moreover, we perform instruction tuning on our IVCR-200K dataset, which contains timestamp-related and natural dialogue data, to further strengthen InterLLaVA's timestamp localization and natural dialogue capabilities.

**Training and Inference.** In training, we implement a two-phase approach. In the first phase, we train a video retrieval model based on the video and text features encoded by BLIP-2(Li et al., 2023a), utilizing X-Pool(Gorti et al., 2022) as the base model. The video retrieval model acts as a plug-in for the multi-modal large language model, retrieving the top-10 video sequences or the top-1 video. In the second phase, we fine-tunes the InterLLaVA with instruction data to achieve instruction following. To better tailor LLaMA for video tasks, we leverage the LoRA(Hu et al., 2021) technique for efficient parameter fine-tuning. To adapt to our IVCR task, we designed a new loss function for training InterLLaVA. For training the large model, we employ a language model loss to generate the target answer $R_a$ with a length of $L_t$. This loss is based on the probability of predicting each word in the answer sequence given the context, such as video tokens $F_v$ and the query tokens $F_q$. It is formulated as

$$
\begin{aligned}
\mathcal{L}_M &= -\log P_\Theta(R_a|F_v, F_q) \\
&= -\sum_{i=1}^{L_t} \log P_\Theta(r_i|R_{a,<i}, F_v, F_q),
\end{aligned}
\tag{1}
$$

where $\Theta$ represents the trainable parameters, and $R_{a,<i}$ refers to the answer tokens preceding the current prediction token $r_i$.

Since our goal is to enhance the large language model's ability for video re-ranking, a direct idea is to directly optimize the predicted video index with the ground truth video index. Let $v_p$ be the predicted video index, and $v_g$ denotes the ground truth video index. The cross-entropy loss function is computed as

$$
\mathcal{L}_V = -\sum_{i=1}^{N} v_{g,i} \log(v_{p,i}),
\tag{2}
$$

where N is the total number of video indices, $v_{g,i}$ is the ground truth probability distribution(with 1 for the correct index and 0 for others), and $v_{p,i}$ is the predicted probability for the i-th video index.

Similarly, let $c_p$ be the predicted video moment interval, and $c_g$ denotes the ground truth video moment interval. we force the model to align each predicted moment candidate with the ground truth moment. Our model is trained by a Intersection over Union (IoU) loss(Yu et al., 2016) as

$$
\mathcal{L}_C = 1 - \text{IoU}(c_p, c_g).
\tag{3}
$$

The overall loss function for training the InterLLaVA is the sum of these three losses, formulated by

$$
\mathcal{L} = \mathcal{L}_M + \alpha \cdot \mathcal{L}_V + \beta \cdot \mathcal{L}_C,
\tag{4}
$$

where $0 \leq \alpha \leq 1$ and $0 \leq \beta \leq 1$ are trade-off parameters that balance the three loss terms.

In inference, we input the textual query into InterLLaVA. Subsequently, InterLLaVA then outputs intent analysis, video prediction or video moment prediction, as well as explainability feedback.

**Table 2: Overall performance comparison of baselines. The "–" indicates not applicable, while bold represents optimal performance.**

| Types | Methods | R@1 ↑ | R@10 ↑ | R@1 IoU=0.5 ↑ | R@1 IoU=0.7 ↑ | BLEU-4 ↑ | GPT-4 Score ↑ |
|---|---|---|---|---|---|---|---|
| Video Retrieval | CLIP4Clip(Luo et al., 2022) | 25.9 | 59.9 | – | – | – | – |
| | X-Pool (Gorti et al., 2022) | 25.3 | 61.1 | – | – | – | – |
| | TS2-Net (Liu et al., 2022b) | 49.1 | 80.1 | – | – | – | – |
| | T-MASS (Wang et al., 2024) | 30.2 | 74.5 | – | – | – | – |
| | BLIP-2 (Li et al., 2023a) | 53.5 | 88.6 | – | – | – | – |
| Moment Retrieval | 2D-TAN Zhang et al. (2020) | – | – | 49.87 | 35.21 | – | – |
| | UMT (Liu et al., 2022a) | – | – | 13.45 | 7.31 | – | – |
| | MMN (Wang et al., 2022b) | – | – | 43.23 | 32.36 | – | – |
| | MomentDiff (Li et al., 2024a) | – | – | 11.59 | 3.4 | – | – |
| | CG-DETR (Moon et al., 2023) | – | – | 48.3 | 28.77 | – | – |
| | GroundingGPT (Li et al., 2024b) | – | – | 12.82 | 4.65 | 0.0018 | 0.68 |
| | VTimeLLM (Huang et al., 2024) | – | – | 17.95 | 7.76 | 0.0035 | 0.74 |
| | TimeChat (Ren et al., 2023) | – | – | 21.24 | 9.80 | 0.0 | 0.64 |
| Interactive Video Retrieval | **InterLLaVA (Ours)** | **58.61** | – | 12.83 | 7.54 | **0.42** | **0.76** |

# 5 EXPERIMENTS

## 5.1 EXPERIMENTAL SETTINGS.

**Datasets Splits.** Our datasets are split into 3 non-overlapping subsets, where 0.8, 0.1 and 0.1 are used for training, validation and testing. Specifically, our training set consists of 11,618 videos and 91,809 textual queries, while the test set includes 449 videos and 2,589 textual queries. The validation set also contains 449 videos and 2,608 textual queries.

**Evaluation Metrics.** We employ two types of metrics to assess our framework. For single-turn evaluation, we utilize R@1 and R@10 to gauge video retrieval proficiency, where 1/10 denotes the top-ranked videos. R@1 IoU={0.5, 0.7} is employed to assess video moment retrieval capability, with IoU=0.5 indicating that the IoU socre between the localized moment and the ground truth must exceed 0.5. Metrics such as BLEU-4 and GPT-4 Score are deployed to evaluate text generation. We classify GPT-4 scores into four categories: highly relevant (1), moderately relevant (0.6), somewhat relevant (0.4), and irrelevant (0). Moreover, we conduct multi-turn performance based on the aforementioned metrics, and any error between between rounds will affect subsequent scores.

**Baselines.** We selected the following five state-of-the-art models as benchmarks for video retrieval, all based on the prevailing pre-trained model CLIP(Radford et al., 2021). Additionally, to comprehensively evaluate the performance of video moment retrieval, we selected five methods as benchmarks. Furthermore, we chose three models based on multi-modal large language models as additional benchmarks for comparison. Please refer to the supplementary materials to obtain the detailed introduction of our baseline.

**Implementation Details.** We employ a pre-trained ViT-G/14 from EVA-CLIP(Sun et al., 2023) and the sliding video Q-Former(Ren et al., 2023) as the image encoder, with LLaMA-2 (7B)(Touvron et al., 2023) as the language model backbone. We train our InterLLaVA using the AdamW optimizer with an initial learning rate of 3e-5 and weight decay of 1e-6 in training phases 1 and 2. Fine-tuning was performed on IVCR-200K for 5 epochs with a batch size of 32. As depicted in Figure 7, the parameters of ViT and LLM remained frozen, while those of the image Q-Former, video Q-Former, and linear layer were tuned. For video retrieval, 12 frames are used, while for moment retrieval, 96 frames are used. All experiments were conducted on 4 Nvidia 4090 GPUs. In addition, the trade-off parameter $\alpha$ an $\beta$ in Eq. (4) are set to 0.01.

## 5.2 OVERALL PERFORMANCE COMPARISON

To evaluate the challenges presented by the IVCR-200K dataset, we conducted a comprehensive study on models for different tasks and our benchmark model. In Table 2, we compared our InterLLaVA with other state-of-the-art methods in video retrieval and video moment methods. Please refer to the supplementary materials to obtain the detailed introduction of our baseline. The detailed introductions to our baselines are provided in supplementary material E.

Table 3: The performance of different pre-retrieval modules.

| Models | R@1 ↑ | R@1 IoU=0.5 ↑ | R@1 IoU=0.7 ↑ |
|---|---|---|---|
| CLIP4Clip | 58.84 | 10.84 | 6.59 |
| X-Pool | 58.61 | 11.18 | 6.15 |
| T-MASS | 57.59 | 11.88 | 6.33 |
| BLIP-2 | 57.91 | 12.83 | 7.54 |

Table 4: Multi-Turn analysis of our framework.

| | R@1 ↑ | R@1 IoU=0.5 ↑ | R@1 IoU=0.7 ↑ |
|---|---|---|---|
| Turn 1# | 41.58 | 6.56 | 5.01 |
| Turn 2# | 15.54 | 9.30 | 5.34 |
| Turn 3# | 10.60 | 9.30 | 5.48 |
| Turn 4# | 6.25 | 12.41 | 8.62 |

**Overall Observations.** 1) Notice that the IVCR task presents significant challenges in the field of video retrieval. While existing traditional models have achieved notable success in single tasks such as video retrieval and video moment retrieval, they fall short compared to our InterLLaVA in terms of considering the importance of flexibly adjusting retrieval strategies based on retrieval intent. This limitation restricts the flexibility and adaptability of video retrieval to some extent. 2) For video moment retrieval, compared to multimodal large language-based methods (e.g., TimeChat(Ren et al., 2023)), traditional methods (e.g., 2D-TAN(Zhang et al., 2020)) achieve superior performance in moment localization. Their advantage lies in the ability to perceive richer contextual information. 3) Moreover, the CLIP-based and BLIP-2-based models, TS2-Net(Liu et al., 2022b) and BLIP-2(Li et al., 2023a), have demonstrated excellent performance on video retrieval task. This proves their ability to more effectively align key textual and video information.

## 5.3 Robustness Analysis

In this section, we will delve into our framework from two perspectives: retrieval module, and multi-turn analysis. We will examine the retrieval module's functionality within the framework, and evaluate the performance of multi-turn dialogue.

**Retrieval Module.** We validate the effectiveness of interactive retrieval modeling by substituting different video retrieval models in Table 3. Our observations are as follows: 1) Upon comparing Tables 2 and 3, it becomes apparent that, for the video retrieval task, CLIP-based models (e.g., X-Pool) demonstrate significantly greater performance improvements ($25.3 \Rightarrow 59.85$) compared to the BLIP-2(Li et al., 2023a) model. 2) In contrast, for the video moment retrieval task, CLIP-based models exhibit slightly diminished performance, suggesting that InterLLaVA's video localization capabilities are influenced by the underlying video retrieval model. Overall, these observations empirically validate the effectiveness of video retrieval models and large language models in modeling interactive retrieval.

**Multi-Turn Analysis.** To evaluate the effectiveness of the model, we compared its performance across different turns of dialogue. As shown in Table 4, as the number of retrieval turns increases, the performance of video retrieval slightly decreases, whereas the performance of video moment retrieval improves.This finding highlights the significant role of context learning in enhancing video localization ability during multi-turn retrieval. It also suggests that video retrieval itself is relatively less influenced by multi-turn context understanding.

## 6 Conclusions

In this paper, we propose a more realistic task to establish an "interaction" between the retrieval system and the user, which involves multi-turn, conversational, and realistic interactions. To facilitate research on this challenging task, we introduce a dataset and framework designed to serve this novel purpose. Notably, our IVCR-200K dataset is a high-quality, bilingual, multi-turn, conversational, and abstract semantic dataset that supports both video and moment retrieval. Our framework is based on MLLMs, which provide more explainable solutions to support users' interaction modes. Our extensive experiments demonstrate the effectiveness of our dataset and framework.

Moving forward, we plan to expand the scope of this research by increasing the size of the dataset and model parameters. Additionally, we will endeavor to develop more sophisticated model architectures to enhance the model's capabilities, considering the challenges posed by interactive retrieval.

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
