These supplemental materials encompass details of investigation of user search behavior feedback (§A), more related works (§B), the details of the dataset (§C), the details of the baselines (§E), more visualization quality (§D), response time of video retrieval (§F), limitations (§G), and quantitative results (§H).

## A  DETAILS OF INVESTIGATION OF USER SEARCH BEHAVIOR FEEDBACK

To verify users' search needs for interactive retrieval, we design a user search behavior feedback survey. It aims to gain in-depth insights into users' real experiences during the search process, identifying potential issues and areas for improvement. The survey covers several aspects: user demographics (such as age, gender, occupation), video search habits, video search usage (whether users prefer to directly search for video moment), evaluation of search effectiveness (including the accuracy and usability of current video search systems, user satisfaction, and expectations for conversational search tools), and issues and improvement suggestions (open-ended questions for users to share problems and suggestions when using interactive search). To better accommodate the habits and preferences of users from different countries, we implemented a diversified survey approach. On the Amazon Mechanical Turk[3] platform, we conducted an online survey specifically designed to target an international audience, successfully engaging 500 participants from various regions. In China, we utilized the Wenjuanxing[4] platform to accurately reach and collect feedback from 500 Chinese users. Additionally, to gather more comprehensive and in-depth data, we organized offline paper surveys. Over the weekends, we collected insights from another 500 participants in high-traffic areas such as university campuses, shopping malls, parks, popular tourist destinations, and subway stations.

## B  MORE RELATED WORK

### B.1  CROSS-MODAL VIDEO RETRIEVAL

**Video Retrieval.** Since TRECVid (Smeaton et al., 2006), the task of cross-modal video retrieval based on text queries has undergone a long development process. The goal of this task is to retrieve relevant videos from a set of video candidates given a text query. In this section, we focus on methods using the deep learning-based paradigm (Chen et al., 2020; Dong et al., 2019; Song et al., 2021; Luo et al., 2022). The basic idea is to encode texts and videos into embeddings, and then learn a common embedding space to do the matching between them. For text encoding, BERT-based models (Devlin et al., 2018) have become the mainstream. While for the video encoding in T2VR, a typical solution has recently evolved from Convolutional Neural Networks (CNN) (e.g., I3D (Carreira & Zisserman, 2017) and 3D ResNets (Hara et al., 2018)) to visual transformers (ViT) (e.g., ViT of CLIP (Radford et al., 2021) and TimeSformer (Bertasius et al., 2021). We can divide related methods into three categories. 1) *Pre-extracted multi-modality fusion* methods (Mithun et al., 2018; Liu et al., 2019; Gabeur et al., 2020; Wang et al., 2021) integrate rich multi-modality information (eg., motion, audio, and face) to improve the performance of T2VR. 2) *Joint text-video pre-training* methods (Wang et al., 2022a; Yan et al., 2023; Li et al., 2022; Ge et al., 2022) train the model with raw video and paired text in an end-to-end manner. 3) *Pre-trained CLIP-based* methods (Luo et al., 2022; Fang et al., 2021; Gorti et al., 2022; Liu et al., 2022b) use CLIP as a text-video backbone and adapt it to T2VR tasks. However, the pre-extracted multi-modality fusion methods are limited by the pre-extracted single-modal features, since these features are not particularly learnt for the target downstream tasks. The joint text-video pre-training methods achieve marginal improvements using a joint text-video pre-training paradigm due to the lack of large-scale text-video datasets. The pre-trained CLIP-based methods largely benefit from the pre-learned vision-text associations inherited frm CLIP and rely on naive mean-pooling or text-conditioned pooling to aggregate visual features.

**Video Moment Retrieval.** As an extension of video retrieval, video moment retrieval task aims to identify specific clips or moments within a video based on a given textual query (Gao et al., 2017; He et al., 2019). Pioneering works have explored various technical avenues, including attention-based retrieval, reinforcement learning, visual-language pretraining, among others. Liu et al. maintain

---

[3]https://www.mturk.com/
[4]https://www.wjx.cn/

a focus on retrieval paradigm, incorporating window segmentation as a preprocessing step, then retrieving moments. (He et al., 2019) devised an agent to pinpoint the start and end timestamps of moments based on reinforcement learning. Following this, some researchers expanded RL-based methods into the spatiotemporal or semantic domains (Cao et al., 2020). Moreover, leveraging pretraining techniques (Zeng et al., 2021; Pan & Zeng, 2023), several prompt-based models have emerged (Zeng, 2022; Liu et al., 2022a), facilitating timestamp prediction through regression.

### B.2 INTERACTIVE RETRIEVAL

The concept of interactive retrieval has long been proposed in the context of combining human-machine learning techniques for multimedia content search. Some studies(Thomee & Lew, 2012; Snoek et al., 2008)interactive have demonstrated that interactive retrieval can significantly improve search performance by enabling users to review search results and refine queries. With the significant progress of deep learning technology in the field of cross-modal video retrieval, interactive video retrieval has re-attracted the attention of researchers. Currently, only a few works(Madasu et al., 2022; Maeoki et al., 2020; Ma & Ngo, 2022; Liang & Albanie, 2023)simple have explored this task. For example, Madasu et al.(Madasu et al., 2022) and Maeoki et al.(Maeoki et al., 2020)adopt a dialogue-based approach, utilizing a series of video-related questions and answers generated by different models as retrieval queries. Meanwhile, addressing the issue that the above methods did not directly involve video question answering, Liang et al.(Liang & Albanie, 2023) employed a video question answering model to generate question-and-answer information in order to improve the retrieval accuracy. Furthermore, Ma et al.(Ma & Ngo, 2022)develop a user simulation for intelligent multimedia applications, leveraging advanced techniques in multimedia content analysis, including concept detection and cross-modal embedding, to enable precise video segment search through human-computer interaction. The aforementioned works are limited to achieving single-task interactive retrieval through methods such as reconstructing retrieval text, using visual question answering models, and simulating users. The technical challenges in modeling multi-turn dialogue retrieval have contributed to the slow development in this direction.

## C  DETAILS OF DATASET

### C.1  DATASET INSTANCE

The instances in our dataset consist of four fields: (1) id: a unique identifier generated using the video name and a random number; (2) type: retrieval intent categorized into seven types—0: chat intent, 1: video retrieval intent, 2: video moment retrieval intent, 3: video to video moment retrieval intent, 4: video moment to video retrieval intent, 5: abstract search intent, 6: analogous search intent, 7: context-independent intent; (3) split: the dataset is divided into training, testing, and validation sets; (4) conversations: multi-turn retrieval formats where "from: human" indicates a query from a human, with the corresponding "value" and "Chinese_value" representing the query content in English and Chinese, respectively. "From: gpt" indicates feedback from GPT, with "gt" representing the retrieved video or video moment—note that "gt_se: [-1,1]" indicates video retrieval, while other values indicate video moment retrieval. Additionally, "video_source" indicates the dataset from which the video is sourced, with the corresponding value providing an interpretable description. Figure 8 shows an example in a unified format. These clearly defined fields allow benchmark users to flexibly construct the necessary training instances and easily evaluate the model.

### C.2  DIVERSITY QUALITY

We conducted an analysis of our video sources, the different types of videos (Figure 9), and performed a frequency analysis of annotated sentences (Figure 10 and 11) to ensure a comprehensive diversity. In addition, as shown in Figure 12, we present the statistics for video retrieval cases.

**Table 5: Comparing InterLLaVA performance across different data distributions**

| N training samples | Category | R@1 ↑ | R@1 IoU=0.5 ↑ | R@1 IoU=0.7 ↑ |
|---|---|---|---|---|
| 2K | Movies | 21.36 | 6.21 | 2.24 |
| 20K | TV shows | 26.73 | 8.24 | 3.98 |
| 1K | ALL | 36.6 | 8.49 | 4.94 |
| 6K | ALL | 45.7 | 11.31 | 5.8 |
| 20K | ALL | 47.96 | 11.52 | 6.45 |
| 60K | ALL | 54.86 | 12.28 | 7.13 |

## C.3 Training Data Distribution Analysis.

Our InterLLaVA model is trained on four different data modes, with the number of training samples ranging from 1K to 60K. Table 5 summarizes the performance evaluation results for all training samples. We observed the following points: 1) In low-sample scenarios, particularly when the sample size is less than 6K, InterLLaVA's accuracy is significantly limited (36.6 vs 54.86 for R@1), showing much lower performance compared to conditions with larger sample sizes. 2) We further explored training the model using samples from a single category (e.g., movies). The experimental results indicate that compared to training on data of the same scale but with more diverse categories, InterLLaVA's video and moment retrieval performance decreased by 21.23% and 3.28% in R@1 and R@1 IoU=0.5, respectively. This result aligns with expectations, as training on more diverse categories allows the model to capture richer features and enhance its generalization ability.

## D MORE VISUALIZATION QUALITY

We present more examples from our IVCR-200K dataset in Figures 13-21.

## E DETAILS OF BASELINES

For video retrieval, we selected the following five state-of-the-art models as benchmarks. We adopt their original setup, using both video and text as model inputs for the video retrieval task. CLIP4Clip(Luo et al., 2022) uses CLIP to extract the frame features and the text features, and then uses the mean pooling to aggregate the feature of all frames for video representation. X-Pool(Gorti et al., 2022) adopts text-conditioned pooling to aggregate visual features. TS2-Net(Liu et al., 2022b) proposes different token shift operations in ViT to learn short-term temporal dependencies across locally adjacent frames. T-MASS(Wang et al., 2024) proposes a stochastic modeling approach to achieve expressive and flexible text embeddings, enhancing the alignment of text and video semantics in the joint space. Furthermore, we reimplement a video retrieval model named BLIP-2, utilizing the video and text features encoded by BLIP-2(Li et al., 2023a), with X-Pool(Gorti et al., 2022) serving as the base model.

For video moment retrieval, we selected six methods as benchmarks. We utilize BLIP-2(Li et al., 2023a) as the encoder to extract video and text features, which are then used as inputs for all models in video moment retrieval. 2D-TAN(Zhang et al., 2020) proposes a novel two-dimensional temporal matrix for moment localization. MMN(Wang et al., 2022b) introduces a mutual matching network that directly models the similarity between language queries and video moments within a joint embedding space. UMT(Liu et al., 2022a) proposes a unified framework for solving joint moment retrieval and highlight detection. CG-DETR(Moon et al., 2023) explores the provision of cues for query-associated video clips within cross-modal attention. MomentDiff(Li et al., 2024a) utilizes diffusion models to diffuse real span to random noise, and then learns to denoise the random noise back to the original span under the guidance of text and video similarity. Moreover, we chose a model based on multi-modal large language models as additional benchmarks. TimeChat(Ren et al., 2023)

**Table 6: Comparing average response times of the model across different test data scales**

| N Testing Samples | Average Response Time of Video Retrieval(s) | Average Response Time of MLLM (s) | Total Average Response Time (s) |
|---|---|---|---|
| 1008 | 0.06 | 0.67 | 0.73 |
| 1534 | 0.06 | 0.59 | 0.65 |
| 1878 | 0.06 | 0.73 | 0.79 |
| 2436 | 0.06 | 0.8 | 0.86 |

proposes a time-sensitive multimodal large language model for long video understanding and precise temporal localization.

## F   RESPONSE TIME OF VIDEO RETRIEVAL

We compared the average response times of video retrieval and Multi-Modal Large Language Model (MLLM) inference, and the results are summarized in the Table 6. Our observations are as follows: 1) The average response time for video retrieval is notably lower compared to the MLLM inference. However, the average response time of the MLLM remains below 1 second, which is acceptable. 2) Testing with different numbers of video samples revealed a slight increase in the average response time for video retrieval. Meanwhile, the average response time of the MLLM remains almost constant, demonstrating that our model does not introduce significant delays even when handling larger volumes of video data.

## G   LIMITATIONS

The IVCR-200K dataset is constrained by the depth of manual annotation and the diversity of real-world data types. It needs to be expanded to cover a wider array of interactive retrieval scenarios, including complex analogy searches, diverse contextual searches, and fine-grained interactive search requirements. Additionally, the current model does not achieve seamless integration of video retrieval and moment retrieval into a unified, efficient end-to-end system. There is considerable potential for improvement in areas such as temporal video modeling, accurate capture of user retrieval intent, and the natural and fluid execution of multi-round dialogues.

## H   QUANTITATIVE RESULTS

Figures22-25 presents a qualitative comparison between InterLLaVA and other video large language models. Our observations are as follows: 1) Video-LLaVA(Lin et al., 2023) has limitations in handling video retrieval, as it is limited to describing the direct content relationship between video and text. It lacks intent analysis for text retrieval and cannot provide interpretable feedback for identifying relevant videos and moment. 2) In contrast, TimeChat(Ren et al., 2023) merely offers the start and end timestamps of videos, lacking any form of feedback on the retrieval outcomes, thus insufficient to cater to users' personalized search demands. 3) InterLLaVA excels at accurately matching the desired retrieval video, precisely locating specific moments within videos, and providing interpretable feedback. However, there is still potential to further enhance the richness and detail of the feedback generated by our model.

```
972
973      {
974          "id": "v_n9eNF1WqFu",
975          "type": [1,2,4], #video-moment-similar
976          "split": "train",
977          "conversations": [
978            {
979              "from": "human",
980              "value": "Person in white clothes are dancing.",
981              "Chinese_value": "穿白色衣服的人在跳舞。",
982              "text_id": "247775f9"
983            },
984            {
985              "from": "gpt",
986              "gt": ["s234d9"],
987              "gt_se": [-1, -1],
988              "video_source": "ActivityNet",
989              "value": "The intent of this query is to find a complete video.",
990              "Chinese_value": "这个查询的意图是找到一个完整的视频。"
991            },
992            {
993              "from": "human",
994              "value": "Locate a video moment of the person spinning in circles.",
995              "Chinese_value": "定位视频中该人转圈的瞬间。",
996              "text_id": "0c1dce9c"
997            },
998            {
999              "from": "gpt",
1000             "gt": ["s234d9"],
1001             "gt_se": [1.3, 5.6],
1002             "video_source": "ActivityNet",
1003             "value": "The intent of this query is to further locate a spinning
1004                 moment.",
1005             "Chinese_value": "这个查询的意图是进一步定位转圈的瞬间。"
1006           },
1007           {
1008             "from": "human",
1009             "value": "Is there any similar moment?",
1010             "Chinese_value": "有没有类似的瞬间？",
1011             "text_id": "5553fa8a"
1012           },
1013           {
1014             "from": "gpt",
1015             "gt": ["45866g"],
1016             "gt_se": [-1, -1],
1017             "video_source": "MSR-VTT",
1018             "value": "The 'similar' in the query may refer to videos of dancing
1019                 in circles.",
1020             "Chinese_value": "查询中的类似可能指的是在圆圈中跳舞的视频。"
1021           }
1022         ]
1023     }
1024
1025
```

**Figure 8: Example JSON of our IVCR-200K dataset.**

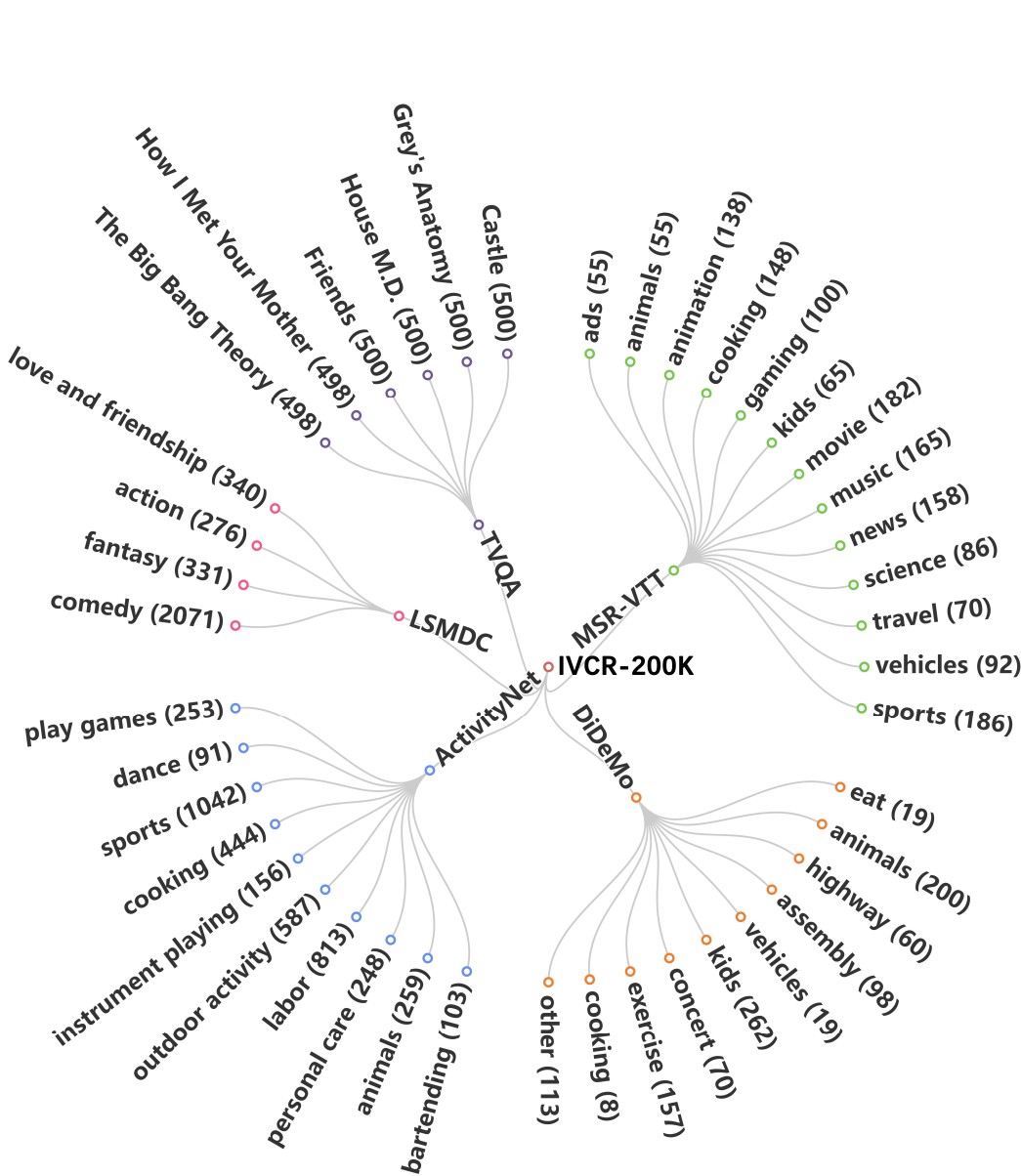

Figure 9: The hierarchical structure of the sources and categories of IVCR-200K.

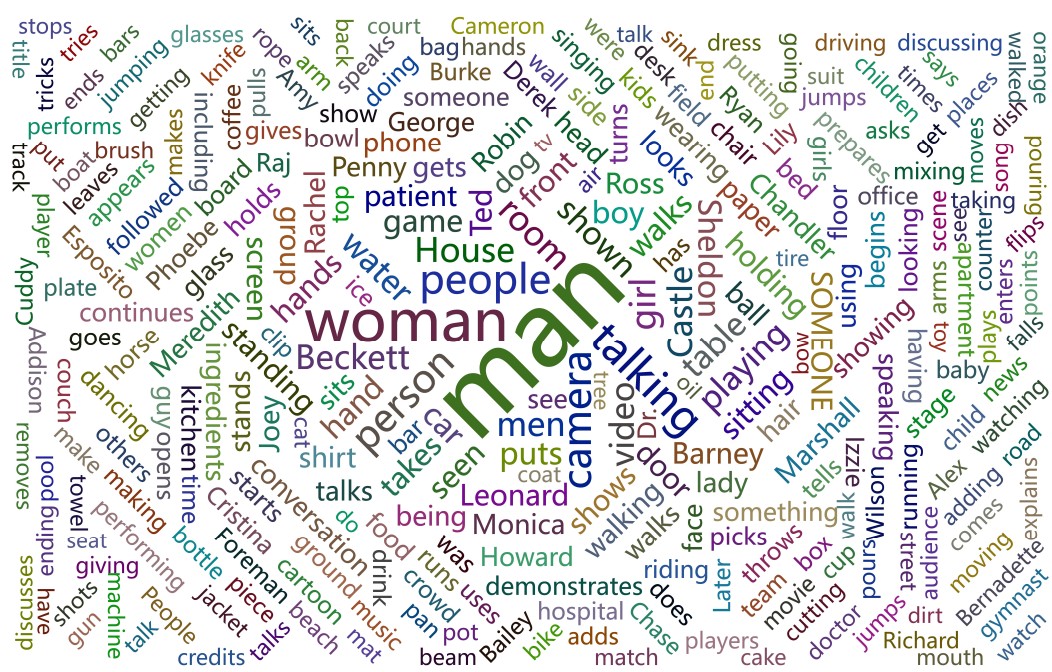

**Figure 10: Word cloud of textual queries from IVCR-200K.**

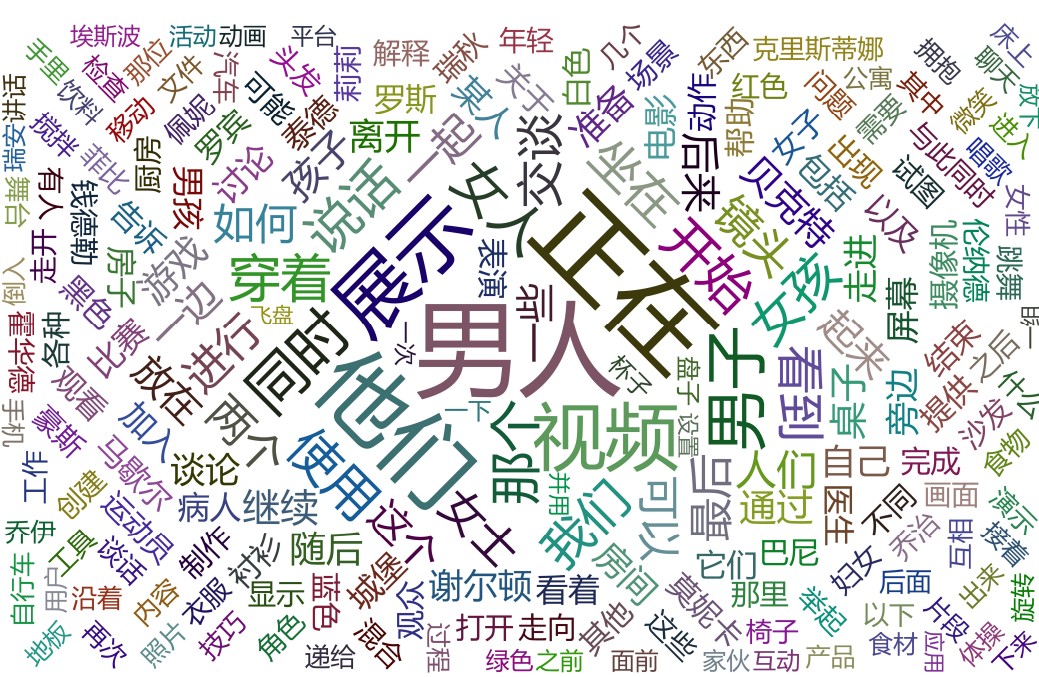

**Figure 11: Chinese Word cloud of textual queries from IVCR-200K.**

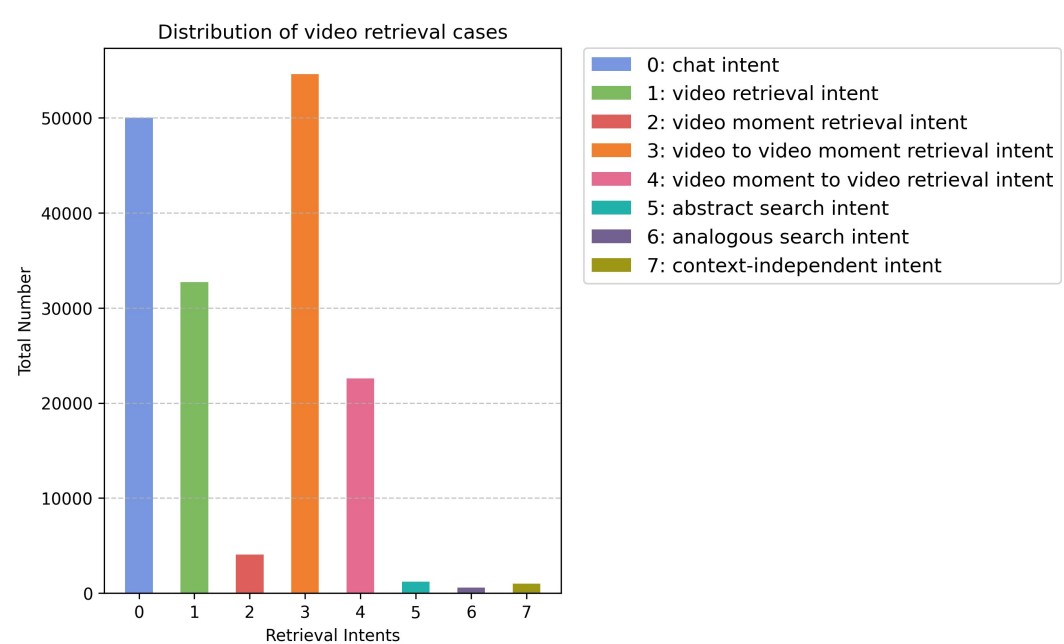

Figure 12: Distribution of video retrieval cases.

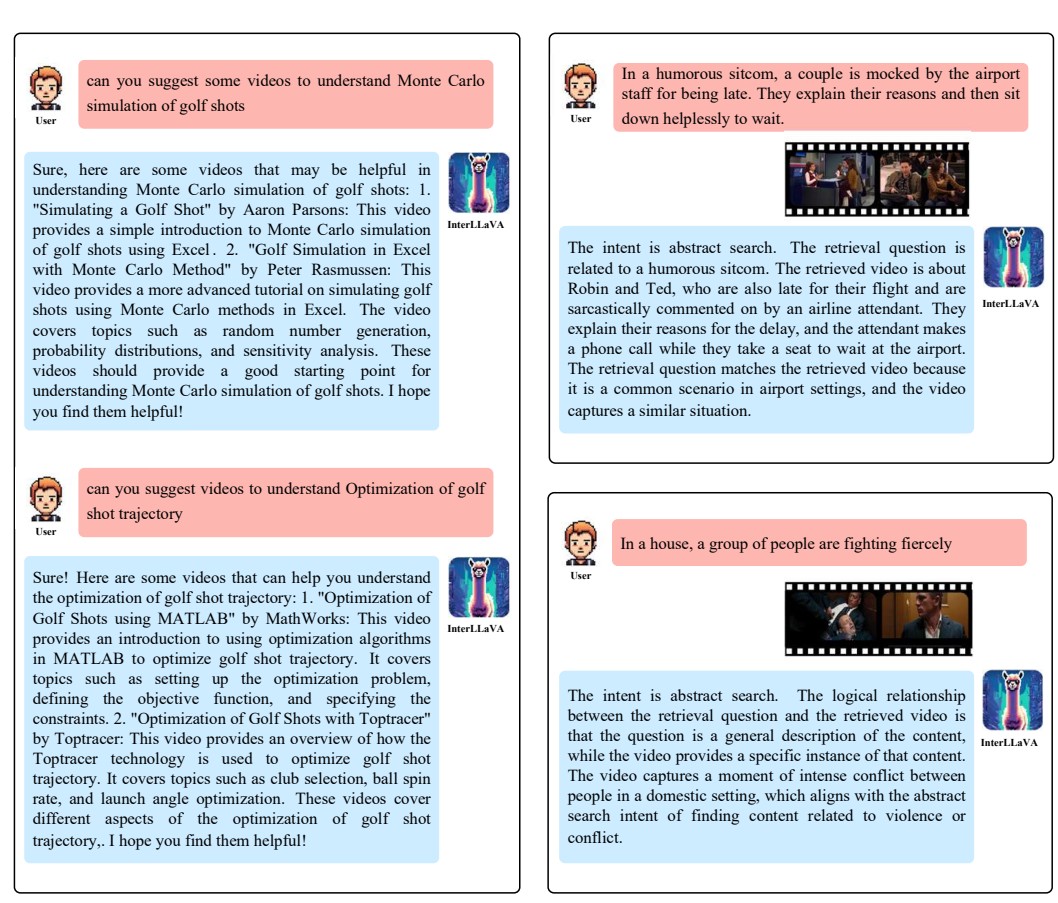

Figure 13: An example of abstract concept retrieval and dialogue from the IVCR-200K dataset.

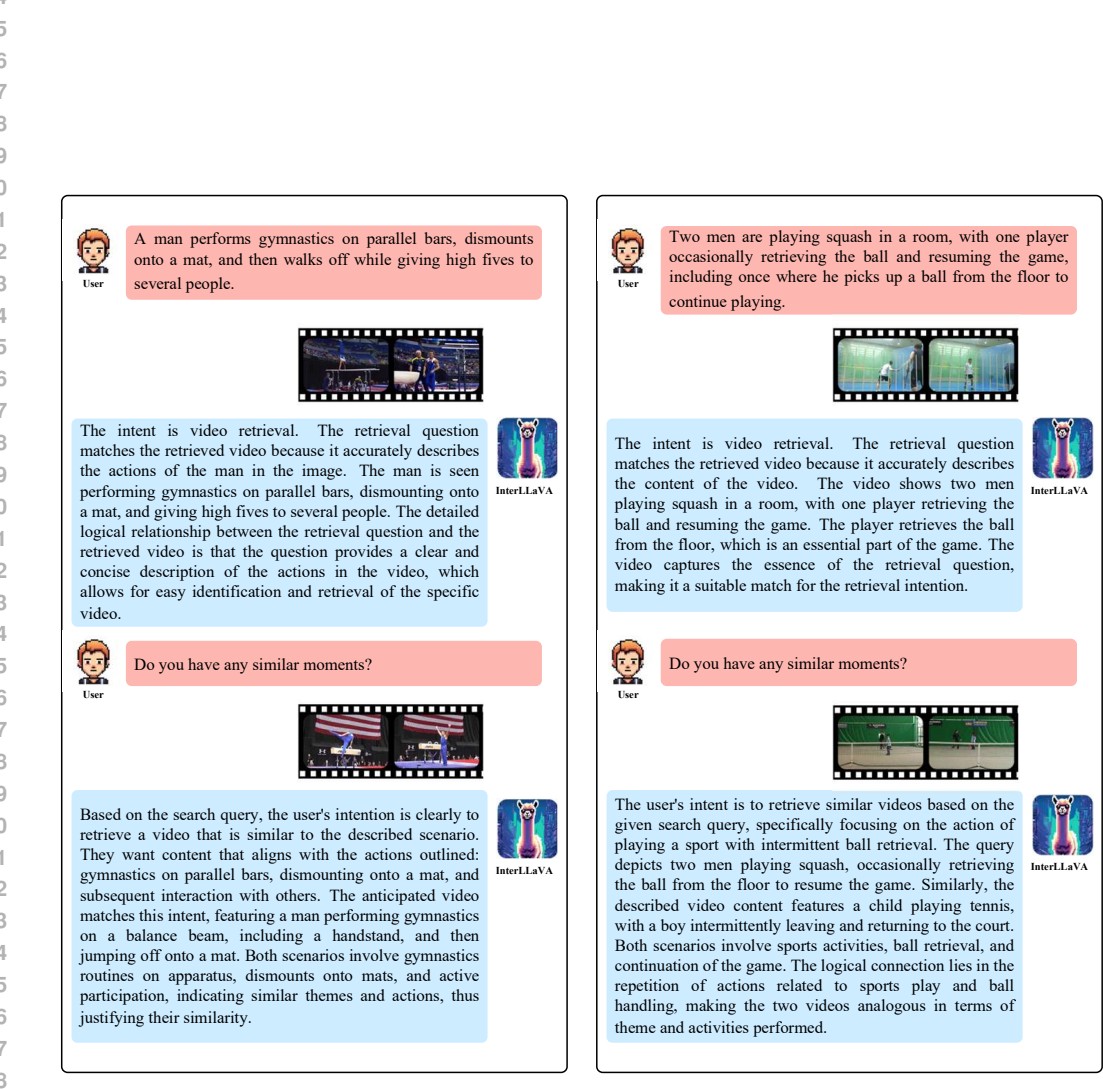

**Figure 14: An example of analogy retrieval from the IVCR-200K dataset.**

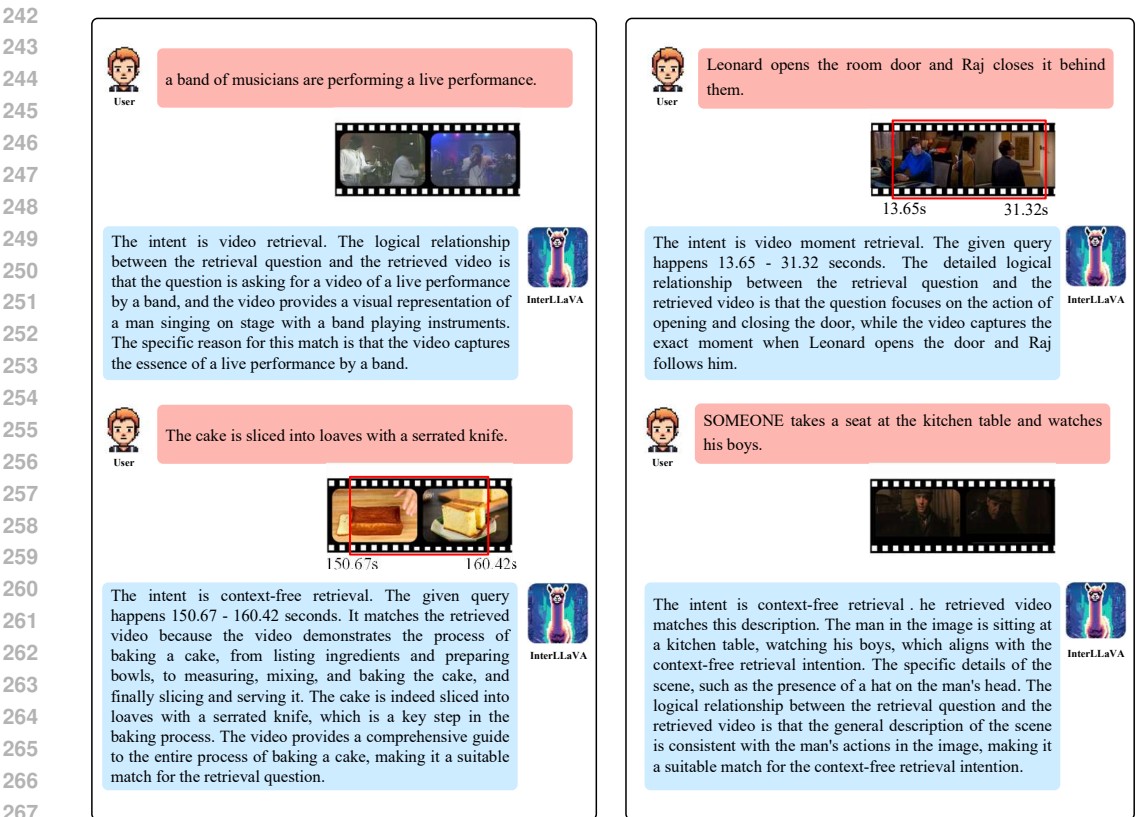

**Figure 15: An example of context-independent retrieval from the IVCR-200K dataset.**

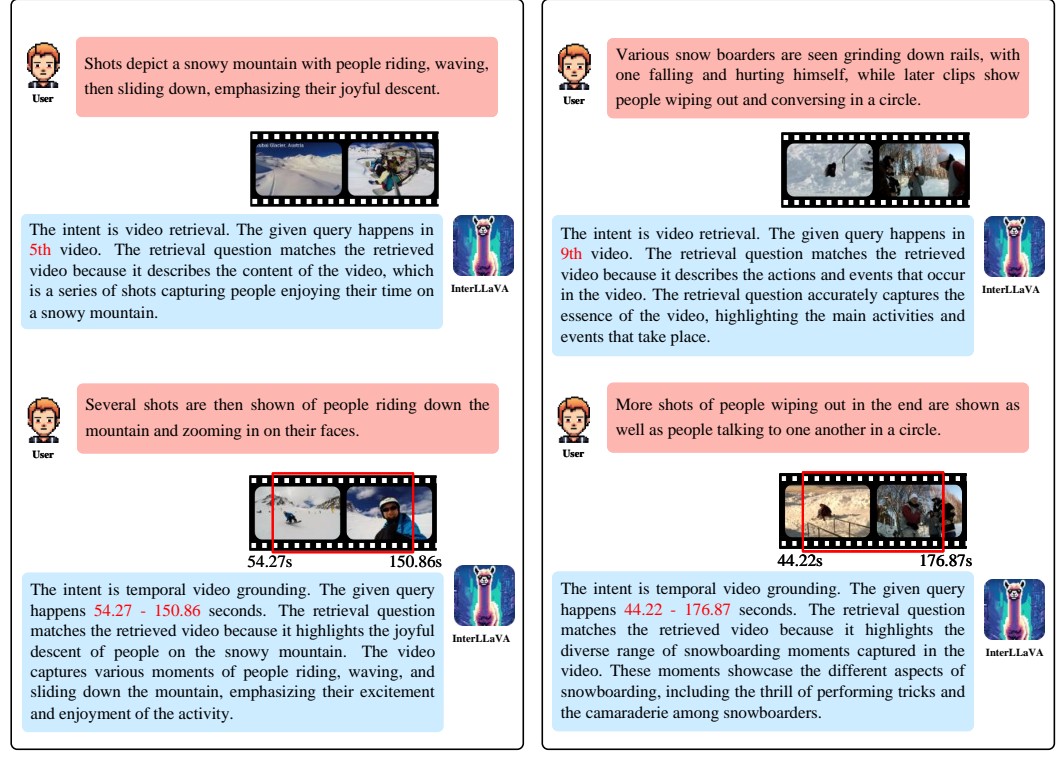

**Figure 16: An example of video-to-video moment retrieval from the IVCR-200K dataset.**

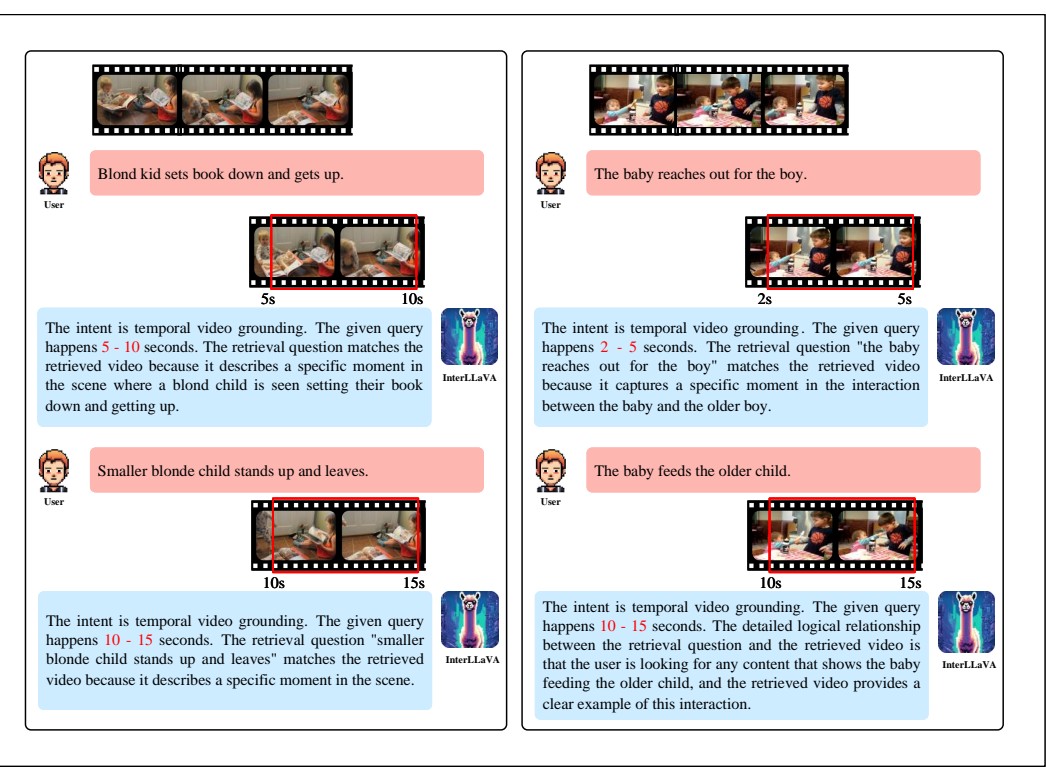

**Figure 17: An example of video-to-video moment retrieval from the IVCR-200K dataset.**

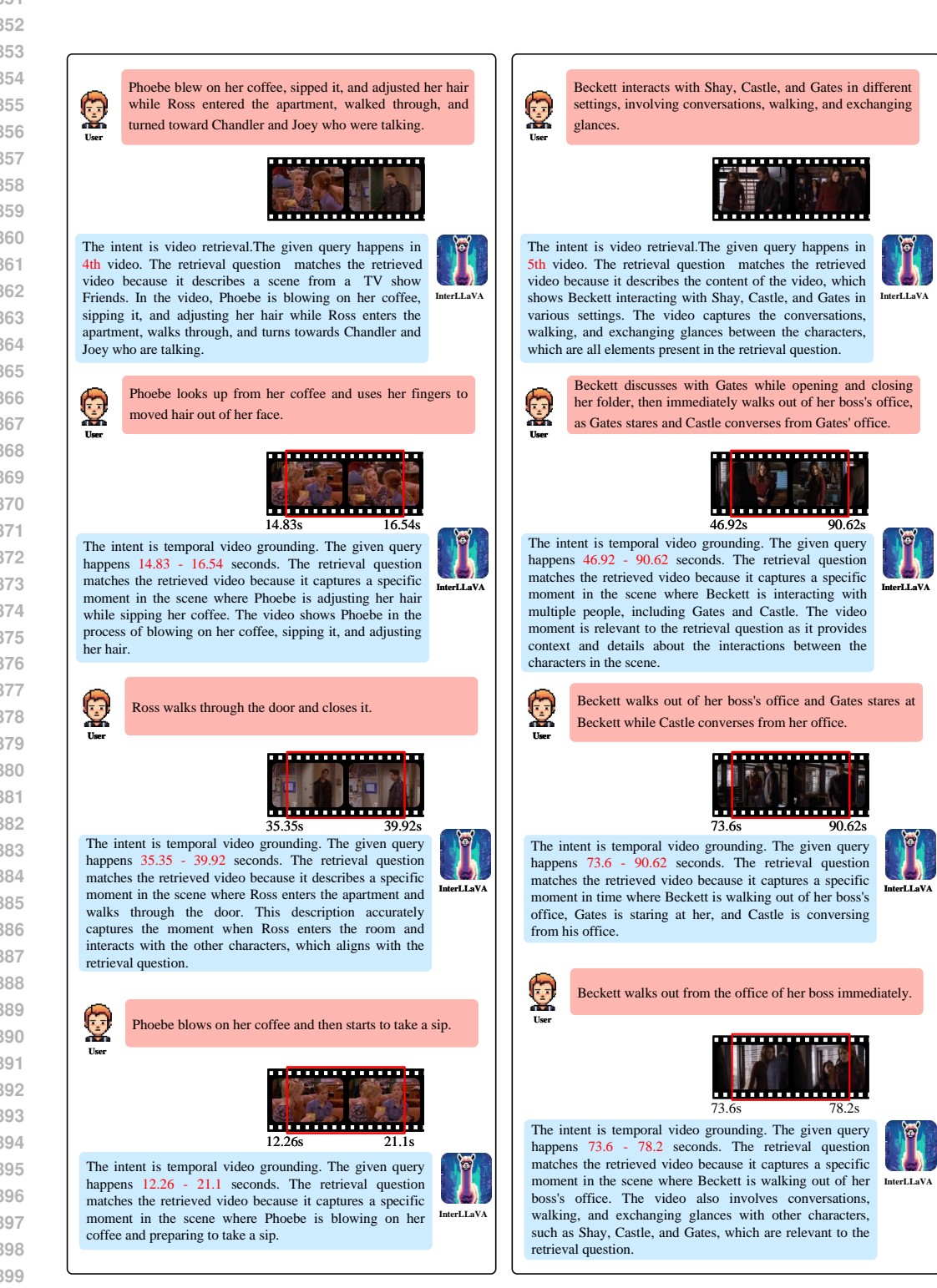

**Figure 18: An example of video-to-video moment retrieval from the IVCR-200K dataset.**

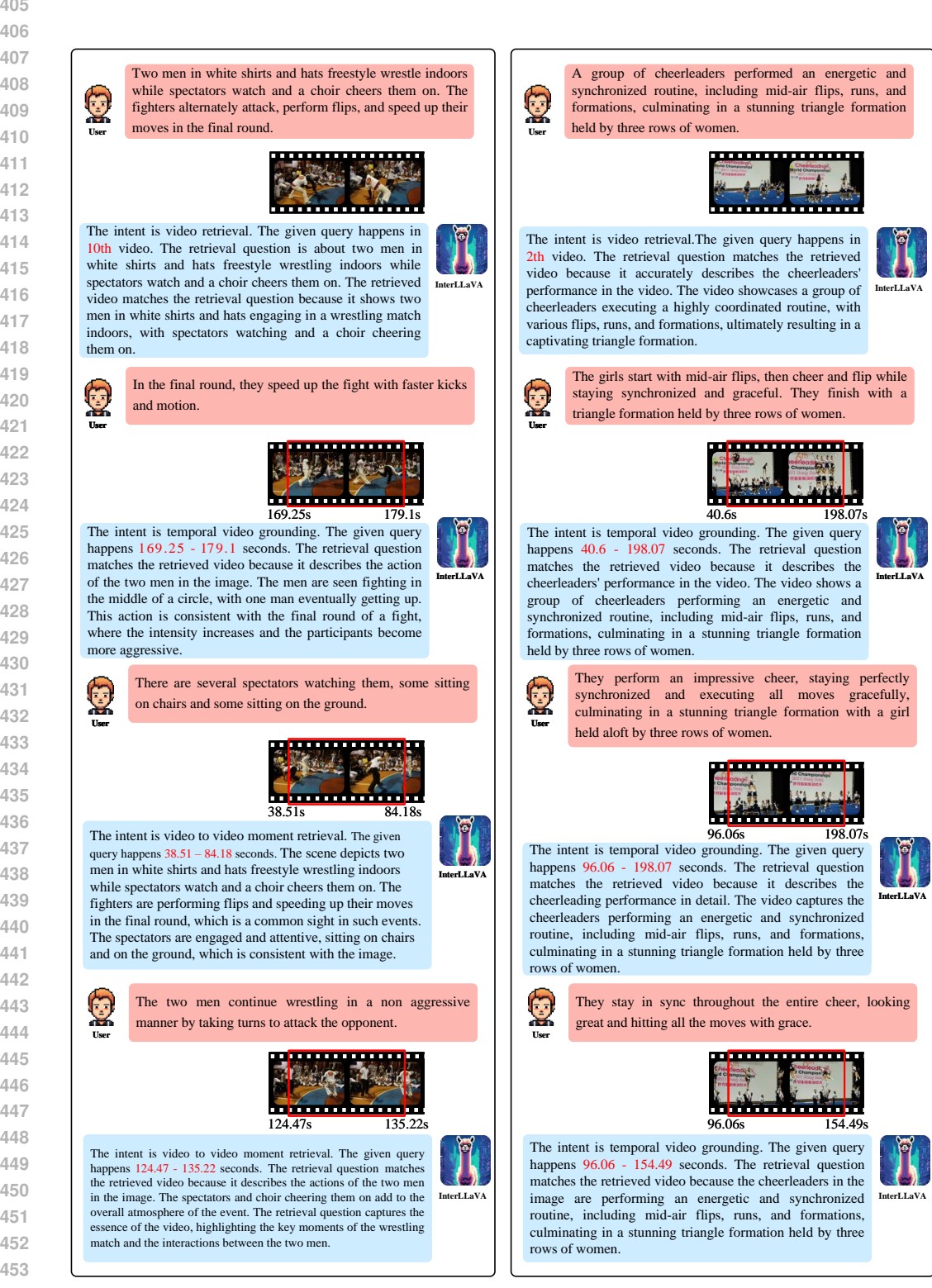

**Figure 19: An example of video-to-video moment retrieval from the IVCR-200K dataset.**

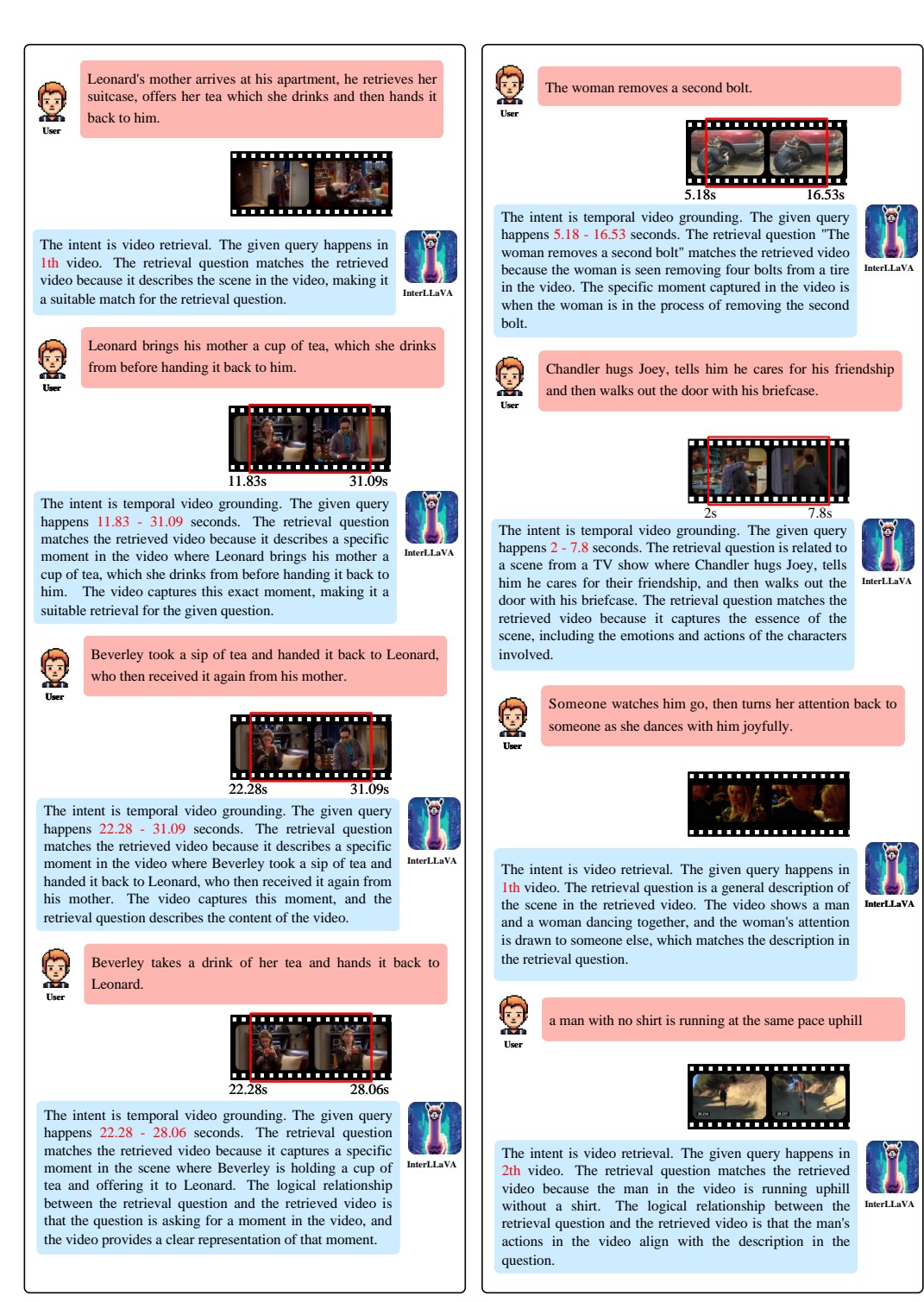

**Figure 20: An example of video-to-video moment retrieval and moment-to-video retrieval from the IVCR-200K dataset.**

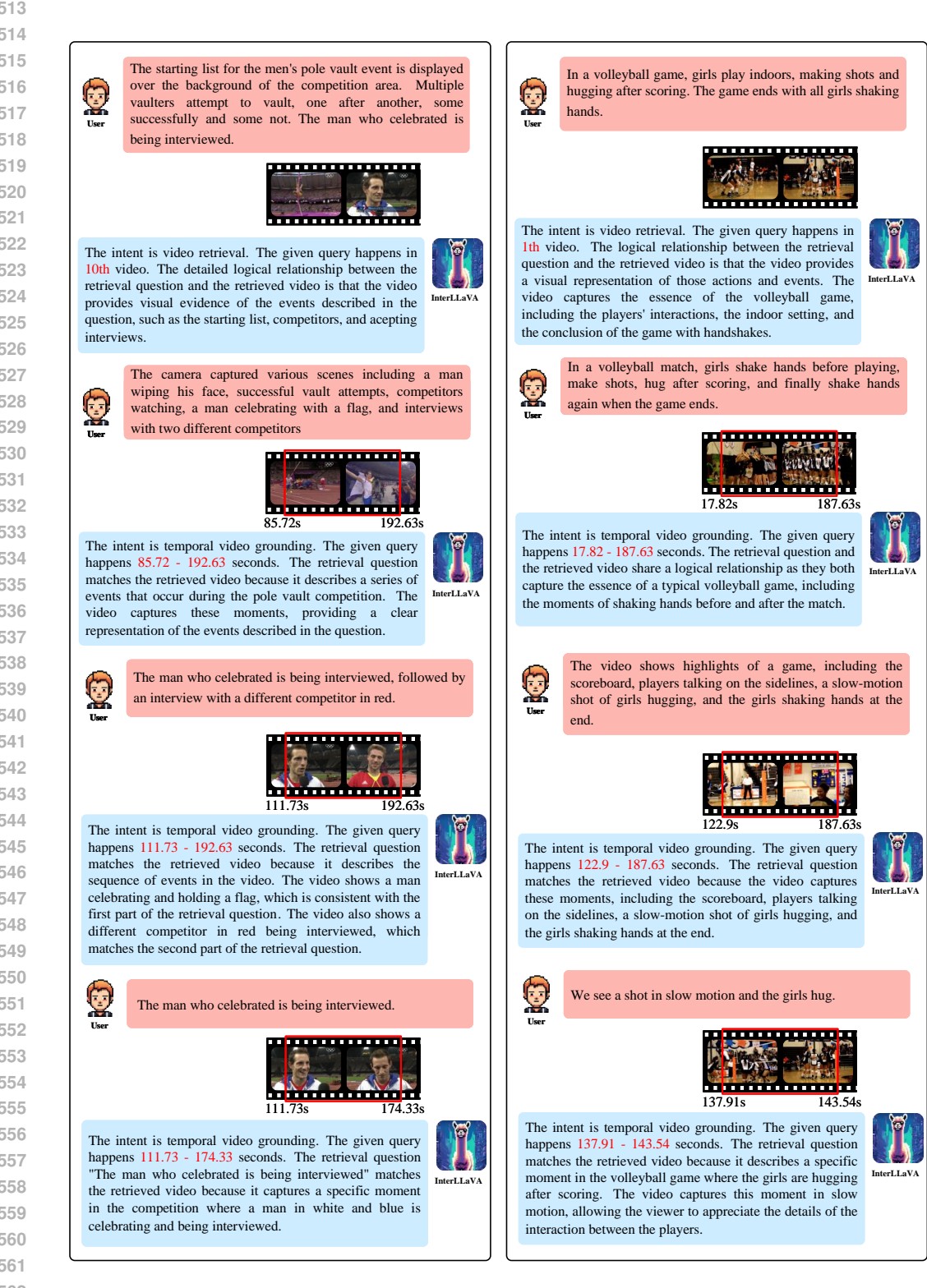

**Figure 21: An example of video-to-video moment retrieval and moment-to-video retrieval from the IVCR-200K dataset.**

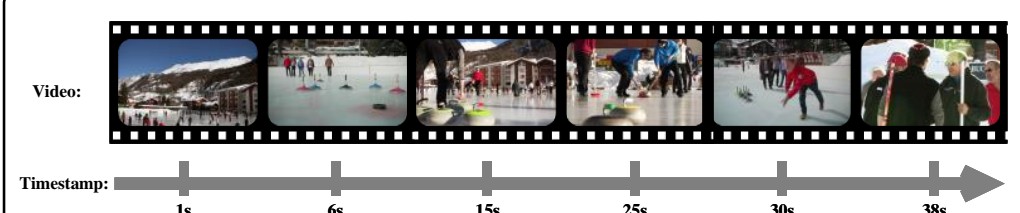

**Video:**

**Timestamp:** 1s    6s    15s    25s    30s    38s

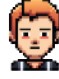

**User**

Query 1#: A large snowy mountain is seen as people gather on the ice. They hit several pucks, sweeping them in a game.

Query 2#: A large snowy mountain is seen as people gather on the ice.

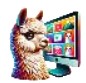

**Video-LLaVA**

Response 1#: The retrieved video matches the retrieval question because it shows a group of people playing a game on ice, hitting pucks, and sweeping them. The logical relationship between the retrieval question and the retrieved video is that the video provides a visual representation of the game being played, which aligns with the description provided in the retrieval question.

Response 2#: The retrieved video matches the retrieval question because it shows a group of people playing curling on an ice rink, which is a winter sport that is often played in snowy mountainous regions. The video also shows the players interacting with each other, which adds to the overall experience of the game.

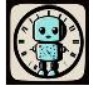

**TimeChat**

Response 1#: ---

Response 2#:The given query happens in the 0.0 - 21.6 seconds.

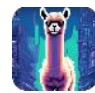

**InterLLaVA**

Response 1#: The intent is video retrieval. The given query happens in 2th video. The retrieval question matches the retrieved video because it describes the scene in the image. The presence of a large snowy mountain in the background adds to the overall context of the scene, making it a fitting description of the video. The logical relationship between the retrieval question and the retrieved video is that the question provides a clear and concise description of the scene, while the video provides the actual visual content that matches the description.

Response 2#: "The intent is temporal video grounding. The given query happens 0.0 - 3.84 seconds. The retrieval question matches the retrieved video because it describes the scene of people engaging in a winter sport on a snowy mountain. The video shows people playing curling, which is a popular winter sport involving sliding stones on a sheet of ice. The presence of a large snowy mountain in the background further supports the idea that this is a winter sports scene. The retrieval question and the retrieved video share a common theme of winter sports and outdoor activities, making them a perfect match.

**Figure 22: Comparing InterLLaVA with other video large language models.**

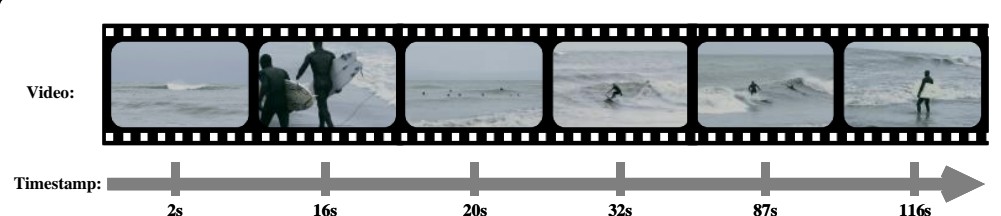

**Timestamp:**

2s      16s      20s      32s      87s      116s

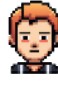

**User**

Query 1#: A large wave moves on the water as people carry surf boards and surf, continuing to ride and walk on the beach.
Query 2#: A large wave is seen moving on the water followed by several people carrying surf boards and surfing along the water.

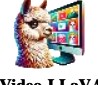

**Video-LLaVA**

Response 1#: The retrieval question matches the retrieved video because it describes the surfers riding the waves and walking on the beach, which are the main activities captured in the video. The video showcases the thrill and excitement of surfing, with the surfers skillfully navigating the waves and the beautiful scenery of the ocean.
Response 2#: The retrieval question matches the retrieved video because the video shows a large wave moving on the water, followed by several people carrying surfboards and surfing along the water. The video provides a detailed visual representation of the surfing activity.

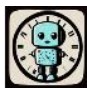

**TimeChat**

Response 1#: ----
Response 2#:The given query happens in 0.0 - 20.8 seconds.、

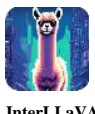

**InterLLaVA**

Response 1#:The intent is video retrieval. The given query happens in 4th video. The retrieval question matches the retrieved video because it describes the scene in the image. The video shows a large wave on the water, with people carrying surf boards and surfing, riding and walking on the beach. The image captures the essence of the retrieval question, making it a suitable match for the video. The logical relationship between the retrieval question and the retrieved video is that the question provides a description of the scene in the image, and the image itself is the actual visual content that matches the description.
Response 2#: The intent is temporal video grounding. The given query happen in 11.62 - 116.98 seconds. The retrieval question matches the retrieved video because it describes the scene of people surfing on the water with large waves. The video captures this moment, showing people surfing and carrying surf boards, which aligns with the description provided in the retrieval question. The logical relationship between the retrieval question and the retrieved video is that the question provides a general description of the scene, while the video offers a specific instance of that scene, showcasing the people surfing and the large waves.

**Figure 23: Comparing InterLLaVA with other video large language models.**

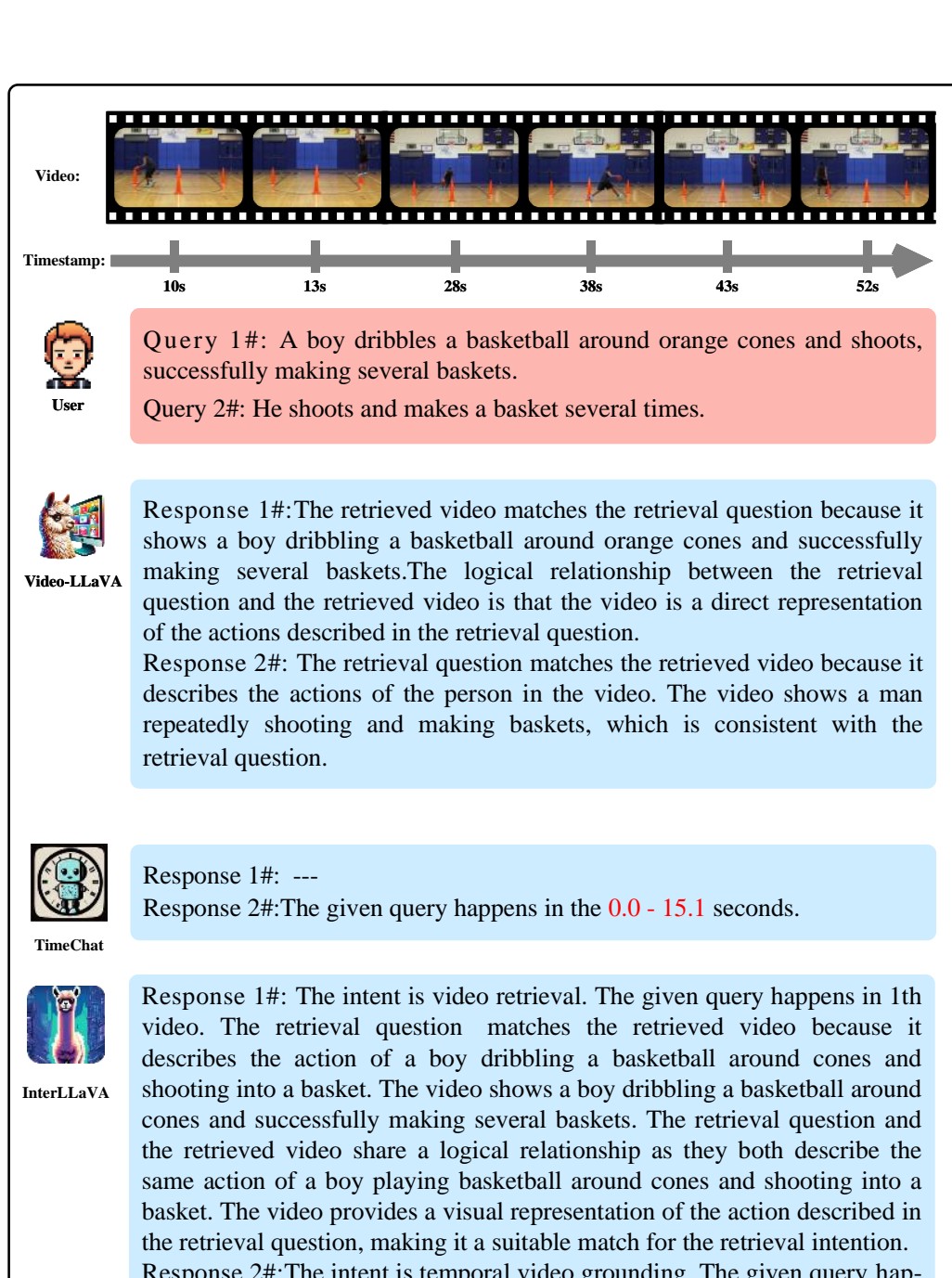

**Figure 24: Comparing InterLLaVA with other video large language models.**

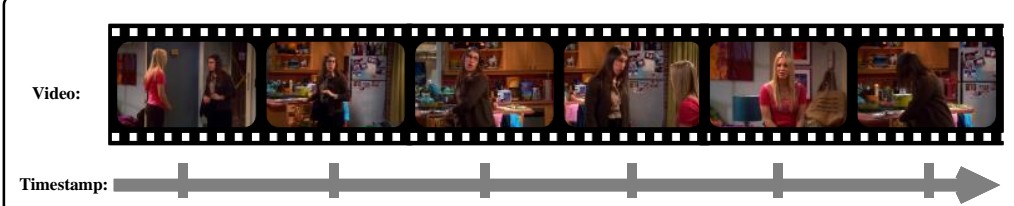

**Video:**

**Timestamp:** 3s    8s    18s    27s    48s    60s

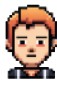

**User**

Query 1#: Amy walks into Penny's apartment, past her while she closes the door, and heads to the kitchen island to take out devices from her bag, while Penny, surprised at Amy's description, sits on the couch arm conversing with her.
Query 2#: Amy walks into Penny's apartment and heads to the kitchen island.

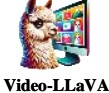

**Video-LLaVA**

Response 1#: The retrieved video matches the retrieval question because it shows Amy walking into Penny's apartment, past her while she closes the door, and heading to the kitchen island to take out devices from her bag. The video also shows Penny, surprised at Amy's description, sitting on the couch arm conversing with her.
Response 2#: The retrieved video matches the retrieval question because it shows Amy walking into Penny's apartment and heading to the kitchen island. The video provides specific details about Amy's actions, such as her walking style, the direction she is heading, and the location she is going to.

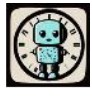

**TimeChat**

Response 1#: ----
Response 2#:The given query happens in 0.0 - 5.9 seconds.

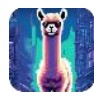

**InterLLaVA**

Response 1#: The intent is video retrieval. The given query happens in 10th video. The retrieved video is a scene from a TV show or movie that matches the retrieval question. The logical relationship between the retrieval question and the retrieved video is that the video shows the exact scenario described in the question. The video captures the moment when Amy walks into Penny's apartment, past her while she closes the door, and heads to the kitchen island to take out devices from her bag. Additionally, the video also shows Penny's reaction to Amy's description, which is her sitting on the couch arm conversing with her.
Response 2#: The intent is temporal video grounding. The given query happens in 43.44 - 49.3 seconds. The retrieval question matches the retrieved video because it captures a specific moment in the scene where Amy is entering Penny's apartment and heading to the kitchen island. This moment is significant as it shows Amy's interaction with Penny and her movement within the apartment. The retrieval question and the retrieved video both focus on the interaction between Amy and Penny, and the video provides a more detailed and accurate representation of the scene.

**Figure 25: Comparing InterLLaVA with other video large language models.**