# OpenReview forum: "IVCR-200K: A Large-Scale Benchmark for Interactive Video Corpus Retrieval"
_ICLR.cc/2025/Conference — ICLR 2025 Conference Withdrawn Submission_

### Official Review · Reviewer_KDhs · 2024-11-02

**Soundness:** 1
**Presentation:** 1
**Contribution:** 2
**Rating:** 3
**Confidence:** 4

**Summary:**

The paper introduces a new dataset, benchmark and method for a new task called Interactive Video Corpus Retrieval. The dataset was curated from existing video datasets and re-framed into a multi-turn retrieval dataset using ChatGPT to connect multiple annotations together to simulate a multi-turn conversation. Based on these videos the authors proposed a benchmark to evaluate performance on video-retrieval and moment-retrieval tasks. For the multi-turn benchmark the authors simply use multiple turn to query the system (not very clear about how this is done) and measure the compounding errors across multiple turns. In terms of the method, the authors propose to solve the problem in a two stage manner, the first one uses a standard retrieval system to rank the top-k videos that are later fed into a Multimodal LLM that re-ranks them and also regress boundaries for the moment-retrieval task. The method performs on-par with other baselines for the retrieval tasks but highly underperforms in the moment-retrieval task (this is using a single turn). When adding multi-turn the method seems to benefit from it for moment-retrieval but seems to be harmful for video-retrieval. Although the reason behind it is unclear since the multi-turn setting is not very clear to me.

**Strengths:**

The paper introduces a method, benchmark and dataset for what seems to be a relevant task for users.
The paper makes an effort on implementing existing methods for the different tasks creating a decent benchmarking of current technologies for video and moment retrieval in the new dataset.

**Weaknesses:**

The paper has several weaknesses that will need to be tackled before being accepted. Most of them are related to clarity of the task,  and dataset, missing references, and clear technical flaws.

1. The paper claims that the 80.8% of the users would prefer interactive search functionality. This is a very strong claim especially with the little information provided on how this number was found. The paper mentions and survey on 500 users, but did not provide the survey. Depending on the nature of the questions and the population the results can be interpreted in different manners. I think the authors need to rephrase of clarify accurately the meaning of this study.

2. The paper does not cite a few important works very related to the topic. First, the ActivityNet dataset was proposed by Caba et al [A], not Krishna, the one referenced in the paper is ActicityNet Captions which is an extension of the original ActivityNet, please give proper attribution to the previous fundamental works in the field. The paper has lots of connection with [B], since the latter one introduced the task of single video moment retrieval in a video corpus. However, there was no mentioned of this paper whatsoever.

3. The paper utilizes automated method to create and refine dataset annotations. They claim that there is a refinement process afterwards but do not describe how is this one performed. Additionally, the multi-turn setting is not explained clearly in the paper. What kind of multi-turn scenarios there are? There are so many way to interact with a system in a multi-turn manner. One could look for videos and refine the search based on the results, look for videos then look for moments in the video, look for moments and then refine the search, etc. The authors mentioned a few of these examples but did not explain how was the benchmark built to cover them. Without this context, it is really hard to understand the results presented in the multi-turn setting in table 4.

4. The authors claim to be measuring BLEU-4 and GPT-4 score metrics on this task. However, I don't understand what kind of answers are the authors evaluating what is the ground-truth for these answers? Is it simply the caption of the video? This requires clarification.


5. The provided baselines make sense for the video and moment retrieval tasks. However, the use of a Multimodal LLM for the fine-grained part of the task (video moment retrieval) seems to be not appropiate. First of all, the MLLM can only process a limited number of frames, which limits the resolution of the system to regress accurate time-stamps. This fact can be seen in table 2, where a method that was state-of-the-art in video-language grounding 4 years ago (2D-TAN) outperforms the proposed method by more that 40 absolute points.

6. The training of the MLLM has one major technical flaw. The paper states that they use the MLLM for re-ranking the top retrieved videos given by the retrieval system. They do it by training the MLLM with three losses, one of them being a cross-entropy loss shown in equation 2. This cross-entropy loss is flawed since it is applied to the video indices in each retrieved set, where the model is expected to identify the "correct" video among the top-k candidates. This setup treats each re-ranking instance as a unique classification problem over a dynamically changing set of "classes" (the retrieved videos), with each new top-k set effectively re-shuffling the class labels. Cross-entropy relies on fixed, stable classes to guide learning, and without this consistency, the model faces moving targets that prevent it from generalizing any meaningful re-ranking ability. A more suitable approach would involve using ranking-specific losses that do not depend on changing class labels but instead focus on optimizing the order of relevance among retrieved videos, such as pairwise or list-wise ranking losses. Given that there was no ablation of this losses, its hard to diagnose how much is this loss affecting the performance of the system. However, if my understanding is correct, this loss is not the way to train a re-ranking system.

7. The conclusions on table 4 are not generalizable to the benchmark, they only talk about the limitations of the current method on performing fine-grained localization. The paper could have proposed a two-stage method using a state-of-the-art retrieval system + a state-of-the-art moment retrieval system and evaluate if under the same setting as table 4. Since this setting is the main selling point of the paper. However, the paper only evaluates the proposed system despite the weaknesses and limitations.



[A] Caba Heilbron, F., Escorcia, V., Ghanem, B., & Carlos Niebles, J. (2015). Activitynet: A large-scale video benchmark for human activity understanding. In Proceedings of the ieee conference on computer vision and pattern recognition (pp. 961-970).


[B] Escorcia, Victor, et al. "Finding moments in video collections using natural language." arXiv preprint arXiv:1907.12763 (2019).

**Questions:**

1. **Interactive Search Functionality**:
   - How was the survey of 500 users conducted? Could the authors provide details on the survey’s methodology, questions, and population demographics?
   - How are the findings interpreted given potential biases in the question design or population selection?
   - Could the authors clarify or rephrase the claim to reflect the limitations of this survey?

2. **Citations and Attribution**:
   - Why was the original ActivityNet dataset attributed to Krishna instead of Caba Heilbron et al.? Could the authors adjust this citation to credit the foundational work correctly?
   - Given the connection to the task introduced in Escorcia et al. (2019), could the authors discuss this paper’s relevance and why it was omitted from the literature review?

3. **Dataset Annotations and Multi-turn Setting**:
   - How is the refinement process (done by humans) for dataset annotations performed? What was the criteria to refine and the methodology used?
   - Could the authors clarify what types of multi-turn interactions were modeled and how these interactions were benchmarked in the dataset?
   - How does the benchmark ensure coverage of diverse multi-turn interaction scenarios, as mentioned in the examples? Authors mentioned that some real multi-turn interactions are included, how were they collected?

4. **Evaluation Metrics (BLEU-4 and GPT-4 Scores)**:
   - What exactly are the authors evaluating with BLEU-4 and GPT-4 scores, and what is considered the ground-truth answer?
   - Are these evaluations solely based on video captions, or are there other elements influencing the ground-truth answers?

5. **Appropriateness of MLLM for Fine-grained Video Moment Retrieval**:
   - How do the authors address the limitations of using an MLLM that processes a limited number of frames, which may impact the system’s ability to regress accurate timestamps?
   - Given that older video-language grounding methods (like 2D-TAN) perform better by over 40 absolute points, is the MLLM genuinely suited for this fine-grained task?

6. **Technical Flaw in MLLM Training with Cross-entropy Loss**:
   - How do the authors justify using cross-entropy loss to re-rank video indices, given that re-ranking each top-k set effectively reshuffles the classes and lacks stable targets?
   - Could the authors consider ranking-specific losses (pairwise or listwise) that avoid dynamic class labels, or provide an ablation study to clarify the cross-entropy loss’s impact on performance?
    - Why cannot the model be evaluated with Recall @ 10 on table 2?

7. **Generalizability of Conclusions on Fine-grained Localization in Table 4**:
   - Would the authors consider implementing a two-stage method combining state-of-the-art retrieval and moment retrieval systems to validate their setting and offer a more comprehensive evaluation? Why was not this considered for table 4?
   - Why were the limitations of the proposed system not benchmarked against alternatives, especially if the multi-turn setting is a main focus of the paper?

---

### Official Review · Reviewer_SPcM · 2024-11-02

**Soundness:** 2
**Presentation:** 2
**Contribution:** 3
**Rating:** 5
**Confidence:** 4

**Summary:**

The focus on this paper is interactive video retrieval. The authors propose a new task formulation, “Interactive Video Corpus Retrieval (IVCR)”, which involves multi-turn interactions between a user and a retrieval system. To study this task, the authors curate videos and captions from 5 existing datasets (TVQA, LSMDC, ActivityNet, DiDeMo, MSR-VTT), and augment these using GPT-4 to construct a new dataset, IVCR-200K. Concretely, GPT-4 is used to re-write the captions/descriptions associated with the original videos, synthesize multi-turn dialogues and predict text-based query responses. The dataset captions are also translated into Chinese.

In order to tackle ICVR, the authors propose an interactive retrieval framework called InterLLaVA. This combines a frozen LLM with a fine-tuned video encoder to perform both video retrieval and video moment retrieval. InterLLaVA is compared with existing methods on IVCR-200K, where it is found to perform strongly for video retrieval, but less well for video moment retrieval.

**Strengths:**

Clarity: Overall, the paper was well-structured. Although I had difficulty understanding some of the specific claims made by the authors (see weaknesses below), I was able to follow the overall message of the work.

Significance: I think interactive video retrieval is a relatively understudied topic with significant commercial potential (indeed, the growing scale of video hosting social media platforms underscores the importance of video search). Consequently, I think the paper focuses on a useful and impactful problem.

Originality: This work explores a different interactive retrieval setting than has been considered in prior work. Specifically, as highlighted in Figure 1(c), the user can mix and match different kinds of queries within a single dialogue. I think this is an interesting and natural direction to explore.

Quality: The authors do a good job of visualizing the contents of the dataset. The hierarchical visualization in Figure 9 of the supplementary material, in particular, conveys the content distribution effectively.

**Weaknesses:**

1. It was good that the authors used a survey to motivate the interactive retrieval task. However, I found it quite difficult to find any details of what the survey contained. I read the description given in Appendix A, which describes the scale of the survey but not the survey questions. Without a detailed description of survey questions and responses statistics, it is quite difficult to assess the validity of claims made in the introduction.  For example, in lines L048-L051, the authors say “our questionnaire indicate that interactive demands exhibit intricate behavioral patterns.”

2. To support the claim that “users desire ‘multi-turn interaction’ with systems” (L044), the authors cite the number of rounds of interaction in ShareGPT conversation dataset as being “remarkably high at 7.27.” (L048).  However, my understanding is that ShareGPT primarily contains data from text-based chat dialogues, rather than multi-turn video retrieval. As such, it doesn’t seem like particularly strong evidence that users want multi-turn behavior in the video setting. A caveat here: unfortunately, the dataset has been taken offline so I’m basing my claim that ShareGPT is primarily text dialogues from my memory - please correct me if I’m mistaken.  The authors could potentially address this by providing additional evidence that is more specific to the multi-turn video retrieval setting, or by presenting arguments for why text-based interactions are meaningful evidence for their claims.

3. Table 1 contains a comparison with prior work. In the “Real interaction” column, only the proposed dataset (ICVR-200K) is ticked. However, if I understand correctly, the dialogue is mostly generated by GPT-4  (using the pipeline shown in Figure 3).  I say mostly, because in L427, the authors write “while most dialogues consist of concatenated single-round exchanges, we also gather a limited number of multi-turn dialogues from real users.” I understand that there is a human expert review process, but I would not describe this data pipeline data as “Real interaction”, given the heavy role played by GPT-4. The proposed pipeline design also makes it somewhat unsurprising that the average length of questions and answers in IVCR-200K is much longer than AVSD (as discussed in L269). (I would expect GPT-4-generated text to have this property.)  It would be appreciated if the authors could clarify how their dataset meets their definition of "Real interaction"?

4. I found several parts of the paper quite difficult to follow. To give a few concrete examples:
(4.1) When defining the “interactive” task in L099-L123, the third component of the definition is “Real interaction.” which is described as follows: “The pioneers create simulated environments to generate interactive data (Ma & Ngo, 2022), but we emphasize that only truly understanding users can optimize a better search experience.” I’m not sure what “truly understanding users” means here? Is this a technical claim or an aspiration? How does it relate to the property of “Real interaction.”  If I should interpret it to mean “using real user data”, would this not imply that real user data should be collected? (My understanding, pointed out in weakness 3, and reflecting Figure 3 of the paper, is that the multi-turn dialogues are mostly generated by GPT-4).
(4.2) In figure 1, it would be helpful to have a caption explain the takeaways. I understand that (a), (b) and (c) each illustrate a different task, but is there some significance to the red sad faces and happy green faces? Some possible interpretations I had were (i) The face is happy for (c) because this is the formulation proposed by the authors; (ii) The face is happy for (c) because this is what the survey suggested that users prefer; (iii) The face is sad for (a) and (b) because the retrieval result quality is lower as a consequence of failing to use the history of previous queries. If the authors could clarify this, it would be appreciated.
(4.3) In the caption for Table 2, it says “bold represents optimal performance.” However, every entry in the “InterLLaVA (Ours)” row is bolded, even when it is not the best performing. For instance, under the R@1 IoU=0.5 metric, this method lags behind several of the Moment Retrieval baselines. Are these baselines comparable? If so, shouldn't 2D-TAN be bold here?

**Questions:**

1. I wasn’t quite sure how to interpret the phrase L427 “while most dialogues consist of concatenated single-round exchanges, we also gather a limited number of multi-turn dialogues from real users.”  Could the authors provide statistics from how much of the multi-turn dialogues are gathered from real users?

2. In Table 2, InterLAVA R@1 accuracy is reported, but R@10 accuracy is not. Was there a rationale for this?

3. In L525, it says “It also suggests that video retrieval itself is relatively less influenced by multi-turn context understanding.” I find this result surprising. Do the authors have intuitions for why the multi-turn setup in Table 4 harms performance on retrieval so dramatically?

---

### Official Review · Reviewer_HXD5 · 2024-11-03

**Soundness:** 3
**Presentation:** 3
**Contribution:** 3
**Rating:** 6
**Confidence:** 5

**Summary:**

Considering the need for personalization and the dynamic requirements of many users, this paper points out that establishing "interaction" between the retrieval system and the user is meaningful. Specifically, it introduces the Interactive Video Corpus Retrieval task (IVCR), which facilitates multi-turn, conversational, and realistic interactions between users and the retrieval system. And then, a large-scale benchmark called IVCR-200K and a comprehensive framework based on multi-modal large language models (MLLMs) are proposed to enhance interaction between models and users.

**Strengths:**

S1. The paper includes a questionnaire survey on user search behavior, highlighting users' preferences for interactive search functionalities. Moreover, it summarizes users’ intricate behavioral patterns that underscore the necessity of an interactive retrieval system.
S2. The proposal of the large-scale benchmark IVCR-200K for interactive video retrieval is a significant contribution, and the paper provides a thorough analysis of its high quality.
S3. The design of the InterLLaVA framework for interactive video retrieval serves as a valuable example for future work in the field.
S4. The writing is well-structured and easy to read, enhancing the paper's overall effectiveness.

**Weaknesses:**

W1. As indicated in Table 2, InterLLaVA's performance in video moment retrieval may not be state-of-the-art, revealing some weaknesses in the model.
W2. During the training of InterLLaVA, questions are divided stage by stage, which may hinder its ability to handle complex or "jumping" questions. This could explain the model's subpar performance in direct video moment retrieval.

**Questions:**

Q1. Can InterLLaVA effectively and directly handle complex or "jumping" questions that involve topic shifts or require nuanced reasoning?
Q2. Figures 4 and 5 indicate that the IVCR-200K dataset exhibits a long-tail distribution, which may affect the model’s learning in interactive video retrieval, particularly its performance on less common queries. Was this factor considered in the construction of IVCR-200K?

---

### Note · Authors · 2024-11-15

I have read and agree with the venue's withdrawal policy on behalf of myself and my co-authors.